Manuscript prepared for Hydrol. Earth Syst. Sci.
with version 2015/11/06 7.99 Copernicus papers of the LaTeX class copernicus.cls.
Date: 18 October 2017

# Precipitation extremes on multiple time scales – Bartlett-Lewis Rectangular Pulse Model and Intensity-Duration-Frequency curves

Christoph Ritschel[1], Uwe Ulbrich[1], Peter Névir[1], and Henning W. Rust[1]

[1]Institut für Meteorologie, Freie Universität Berlin, Carl-Heinrich-Becker-Weg 6-10, D-12165 Berlin, Germany

*Correspondence to:* Christoph Ritschel (christoph.ritschel@met.fu-berlin.de)

**Abstract.** For several hydrological modelling tasks, precipitation time series with a high (i.e. sub-daily) resolution are indispensable. This data is, however, not always available and thus model simulations are used to compensate. A canonical class of stochastic models for sub-daily precipitation are Poisson-cluster processes, with the original Bartlett-Lewis (OBL) model as a prominent representative. The OBL model has been shown to well reproduce certain characteristics found in observations. Our focus is on intensity-duration-frequency relationship (IDF), which are of particular interest in risk assessment. Based on a high resolution precipitation time series (5-min) from Berlin-Dahlem, OBL model parameters are estimated and IDF curves are obtained on the one hand directly from the observations and on the other hand from OBL model simulations. Comparing the resulting IDF curves suggests that the OBL model is able to reproduce main features of IDF statistics across several durations but cannot capture rare events (here an event with a return period larger than 1000 years on the hourly time scale). In this paper, IDF curves are estimated based on a parametric model for the duration dependence of the scale parameter in the Generalised Extreme Value distribution; this allows to obtain a consistent set of curves over all durations. We use the OBL model to investigate the validity of this approach based on simulated long time series.

## 1 Introduction

Precipitation is one of the most important atmospheric variables. Large variations on spatial and temporal scales are observed, i.e. from localised thunderstorms lasting a few tens of minutes up to mesoscale hurricanes lasting for days. Precipitation on every scale affects everyday life: short but intense extreme precipitation events challenge the drainage infrastructure in urban areas or might put agricultural yields at risk; long-lasting extremes can lead to flooding (Merz et al., 2014). Both, short intense and long-lasting large-scale rainfall can lead to costly damages, e.g. the floodings in Germany in 2002 and 2013 (Merz et al., 2014), and are therefore the object of much research.

Risk quantification is based on an estimated frequency of occurrence for events of a given intensity and duration. This information is typically summarised in an Intensity-Duration-Frequency (IDF) relationship (e.g., Koutsoyiannis et al., 1998), also referred to as IDF curves. These curves are typically estimated from long observed precipitation time series, mostly with a sub-daily resolution to include also short durations in the IDF relationship. These are indispensable for some hydrological applications, e.g., extreme precipitation characteristics are derived from IDF curves for planning, design and operation of drainage systems, reservoirs and other hydrological structures. One way to obtain IDF curves is modelling block-maxima for a fixed duration with the generalised extreme value distribution (GEV, e.g., Coles, 2001). Here, we employ a parametric extension to the GEV which allows a simultaneous modelling of extreme precipitation for all durations (Koutsoyiannis et al., 1998; Soltyk et al., 2014).

Due to a limited availability of observed high-resolution records with adequate length, simulations with stochastic precipitation models are used to generate series for subsequent studies (e.g., Khaliq and Cunnane, 1996; Smithers et al., 2002; Vandenberghe et al., 2011). The advantages of stochastic models are their comparably simple formulation and their low computational costs allowing to quickly generate large ensembles of long precipitation time series. A review of these models is given in, e.g., Onof et al. (2000) and Wheater et al. (2005). A canonical class of sub-daily stochastic precipitation models are Poisson cluster models with the original Bartlett-Lewis (OBL) model as a prominent representative (Rodriguez-Iturbe et al., 1987, 1988; Onof and Wheater, 1994b; Wheater et al., 2005). The OBL model has been shown to be able to well reproduce certain characteristics found in precipitation observations (Rodriguez-Iturbe et al., 1987).

Due to the high degree of simplifications of the precipitation process, known drawbacks of the OBL model include the inability to reproduce the proportion of dry periods as reported by Rodriguez-Iturbe et al. (1988) and Onof (1992), and underestimation of extremes as found by, e.g. Verhoest et al. (1997) and Cameron et al. (2000), especially for shorter durations. Futhermore, problems occur for return levels with associated periods longer than the time series used for calibrating the model (Onof and Wheater, 1993). Several extensions and improvements to the model have been made. Rodriguez-Iturbe et al. (1988) introduced the randomised parameter Bartlett-Lewis model, allowing for different types of cells. Improvements in reproducing the probability of zero rainfall and capturing extremes have been shown for this model (Velghe et al., 1994). A gamma-distributed intensity parameter and a jitter were introduced by Onof and Wheater (1994b) for more realistic irregular cell intensities. Nevertheless, problems still remain as Verhoest et al. (2010) discussed the occurrence of infeasible (extremely long lasting) cells and a too severe clustering of rain events was found by Vandenberghe et al. (2011). Including third-order moments in the parameter estimation showed an improvement in the Neyman-Scott model's extremes (Cowpertwait, 1998). For the Bartlett-Lewis variant Kaczmarska et al. (2014) found that a randomised parameter model shows no improvement in fit compared to the OBL model in which the skewness was included in the parameter estimation.

Furthermore, an inverse dependence between rainfall intensity and cell duration showed improved performance, especially for extremes at short time scales (Kaczmarska et al., 2014). Here, we focus on the OBL model with and without the third-order moment included. This model is still part of a well-established class of precipitation models and the reduced complexity is appealing as it allows

to be used in a non-stationary context (Kaczmarska et al., 2015).

The OBL model and intensity-duration-frequency relationships are of particular interest to hydrological modelling and impact assessment. In the following, we address the three research questions by means of a case study: 1) Is the OBL model able to reproduce the intensity-duration relationship found in observations?, 2) How are IDF curves affected by very rare extreme events which are

unlikely to be reproduced with the OBL model in a reasonably long simulation? and 3) is the parametric extension to the GEV a valid approach to obtain IDF curves? For the class of multi-fractal rainfall models, question 1) has been addressed by Langousis and Veneziano (2007).

Section 2 gives a short overview of the OBL model used here, Sect. 3 briefly explains IDF curves and how to obtain them from precipitation time series. This is followed by a description of the data

used (Sect. 4). The results section (Sect.5) starts with a description of the estimated OBL model parameters for the case study area (Sect. 5.1), discusses then the ability of this model to reproduce IDF curves (Sect. 5.2) and the influence of a rare extreme (Sect. 5.3) and closes with a comparison of the duration dependent GEV approach to IDF curves with individual duration GEV quantiles (Sect. 5.4). Section 6 discusses results and concludes the paper.

## 2   Bartlett-Lewis rectangular pulse model

From early radar based observations of precipitation, a hierarchy of spatio-temporal structures was suggested by Pattison (1956) and Austin and Houze (1972): intense rain structures (denoted as cells) tend to form in the vicinity of existing cells and thus cluster in larger structures, so called storms or cell clusters. Occurrence of these cells and storms (cell clusters) can be described using Poisson-

processes. This makes the Poisson-cluster process a natural approach to stochastic precipitation modelling.

The idea of modelling rainfall with stochastic models exists since Le Cam (1961) modelled rain gauge data with a Poisson-cluster process. Later, Waymire et al. (1984) and Rodriguez-Iturbe et al. (1987) continued the development of this type of model. Poisson-cluster rainfall models are charac-

terised by a hierarchy of two layers of Poisson processes.

Similarly to various others studies (e.g., Onof and Wheater, 1994a; Kaczmarska, 2011; Kaczmarska et al., 2015), we choose the original Bartlett-Lewis (OBL) model, a popular representative of Poisson-cluster models with a set of five physically interpretable parameters (Rodriguez-Iturbe et al., 1987). At the first level cell clusters (storms) are generated according to a Poisson-process with

a cluster generation rate $\lambda$ and an exponentially distributed life-time with expectation $1/\gamma$. Within

each cell cluster, cells are generated according to a Poisson process with cell generation rate $\beta$ and exponentially distributed life-time with expectation $1/\eta$, hence the term Poisson-cluster process. Associated with each cell is a precipitation intensity being constant during cell life-time, exponentially distributed with mean $\mu_x$. The constant precipitation intensity in one cell gave rise to the name rectangular pulse. Model parameters are summarised in the parameter vector $\boldsymbol{\theta} = \{\lambda, \gamma, \beta, \eta, \mu_x\}$.

Simulations with the OBL model are in continuous-time on the level of storms and cells. We aggregate the resulting cell rainfall series to hourly time series. Figure 1 shows a sketch of the OBL model illustrating the two levels: start of cell-clusters (storms) are shown as red dots, within each cluster, cells (blue rectangular pulses) are generated during the clusters' life-time starting from the cluster origin. The cells' lifetime is shown as horizontal and their precipitation intensity as vertical extension in Fig. 1; cells can overlap. This continuous-time model yields a sequence of pulses (cells) with associated intensity, see Fig. 2 (top panel). Adding up the intensity of overlapping pulses yields a continuous-time step-function (Fig. 2, middle panel). Although time continuous, this function is not continuously differentiable in time due to the rectangular pulses. Observational time series are typically also not continuously differentiable as they are discretised in time. Summing up the resulting continuous series for discrete time intervals makes it comparable with observations and renders unimportant the artificial jumps from the rectangular pulses present in the continuous series, Fig. 2 (bottom layer).

An alternative to the Bartlett-Lewis process is the Neyman-Scott process (Neyman and Scott, 1952). The latter is motivated from observations of the distribution of galaxies in space. In the Neyman-Scott process cells are distributed around the centre of a cell cluster. Both are prototypical models for sub-daily rainfall and are discussed in more detail in Wheater et al. (2005).

Due to known drawbacks of the OBL model, several improvements and extensions have been made in the past: Rodriguez-Iturbe et al. (1988) introduced the random parameter model, allowing for different types of cells, and additionally Onof and Wheater (1994b) used a jitter and a gamma-distributed intensity parameter to account for a more realistic irregular shape of the cells. Cowpertwait et al. (2007) improved the representation of sub-hourly time scales by adding a third layer, pulses, to the model. Non-stationarity has been addressed by Salim and Pawitan (2003) and Kaczmarska et al. (2015). Applications of these kind of models include the implementing of copulas to investigate wet and dry extremes (Vandenberghe et al., 2011; Pham et al., 2013), regionalisation (Cowpertwait et al., 1996a, b; Kim et al., 2013) and accounting for interannual variability (Kim et al., 2014).

Parameter estimation for the OBL model is by far not trivial. The canonical approach is a method-of-moment-based estimation (Rodriguez-Iturbe et al., 1987) using the objective function

$$Z(\boldsymbol{\theta}; \mathbf{M}) = \sum_{i=1}^{k} w_i \left[ 1 - \frac{\tau_i(\boldsymbol{\theta})}{M_i} \right]^2 . \tag{1}$$

This function relates moments of precipitation sums $\tau_i(\boldsymbol{\theta})$ derived from the model with parameters $\boldsymbol{\theta}$ to empirical moments $M_i$ from the time series. The set of $k$ moments $M_i$ is typically chosen from the first and second moments obtained for different durations h. Here, we use the mean at one hour aggregation time, the variances, the lag-1 auto-covariance function and the probability of zero rainfall for h $\in \{1\,\text{hour}, 3\,\text{hours}, 12\,\text{hours and } 24\,\text{hours}\}$, similar to Kim et al. (2013), and thus end up with $k = 13$ moments $M_i$. Their analytic counterparts $\tau_i(\boldsymbol{\theta})$ are derived from the model. The weights in the objective function were chosen to be $w_{i=1} = 100$ and $w_{i \neq 1} = 1$, similar to Cowpertwait et al. (1996a), emphasising the first moment $M_1$ (mean) 100 times more than the other moments.

It turns out that $Z(\boldsymbol{\theta}; \mathbf{M})$ has multiple local minima and optimisation is not straightforward. To avoid local minima, the optimisation is repeated many times with different initial guesses for the parameters sampled from a range of feasible values in parameter space using the Latin-Hypercube sampling algorithm (McKay et al., 1979). As only positive model parameters are meaningful, optimisation is performed on log-transformed parameters. Similarly to Cowpertwait et al. (2007), we use a symmetric objective function

$$Z(\boldsymbol{\theta}; \mathbf{M}) = \sum_{i=1}^{k} w_i \left\{ \left[ 1 - \frac{\tau_i(\boldsymbol{\theta})}{M_i} \right]^2 + \left[ 1 - \frac{M_i}{\tau_i(\boldsymbol{\theta})} \right]^2 \right\}. \tag{2}$$

A few tests indicate that the symmetric version is robust and faster in the sense that fewer iterations are needed to ensure convergence into the global minimum (not shown). Numerical optimisation techniques based on gradient calculations, e.g. *Nelder-Mead* (Nelder and Mead, 1965) or *BFGS* (Broyden, 1970; Fletcher, 1970; Goldfarb, 1970; Shanno, 1970), are typically used. For the current study, we use R's `optim()` function choosing *L-BFGS-B* as the underlying optimisation algorithm (R Core Team, 2016) and 100 different sets of initial guesses for the parameters sampled using the Latin-Hypercube algorithm.

Following studies by Cowpertwait (1998) and Kaczmarska et al. (2014), we include the third moment in the parameter estimation using analytical expressions derived by Wheater et al. (2006), replacing the probability of zero rainfall in the objective function. Thus, still 13 moments are used to calibrate the OBL model. Due to comparability with other studies most of our analyses will not include the third moment though. A comparison between IDF curves of the model calibrated with the third moment and with the probability of zero rainfall will be carried out, to discuss the effect of including the third moment.

Models of this type suffer from parameter non-identifiability, meaning that qualitatively different sets of parameters lead to minima of the objective function with comparable values (Verhoest et al., 1997). A more detailed view on global optimisation techniques and comparisons between different objective functions is given in Vanhaute et al. (2012).

During this work the authors developed and published the R-package `BLRPM` (Ritschel, 2017). The package includes functions for simulation and parameter estimation.

## 3 Intensity-Duration-Frequency

Intensity-duration-frequency (IDF) curves show *return levels* (intensities) for given *return periods* (inverse of frequencies) as a function of *rainfall duration*. Their formulation goes back to Bernard (1932). They are frequently used for supporting infrastructure risk assessment (e.g., Simonovic and Peck, 2009; Cheng and AghaKouchak, 2013). IDF curves are an extension to classical extreme value statistics. The latter aims at better characterising the tails of a distribution by using parametric models derived from limit theorems (e.g., Embrechts et al., 1997). There are two main approaches: modelling block-maxima (e.g., maxima out of monthly or annual blocks) with the generalised extreme value distribution (GEV) or modelling threshold excesses with the generalised Pareto distribution (GPD) (e.g., Coles, 2001; Embrechts et al., 1997). We choose the block-maxima approach with the general extreme value distribution

$$G(z) = \exp\left\{ -\left[1 + \xi\left(\frac{z-\mu}{\sigma}\right)\right]^{-\frac{1}{\xi}} \right\} \tag{3}$$

as parametric model for the block-maxima $z$. The GEV is characterised by the location parameter $\mu$, the scale parameter $\sigma$ and the shape parameter $\xi$. These can be estimated from block-maxima using a maximum-likelihood estimator (e.g., Coles, 2001). Here, we use maxima from monthly blocks. To avoid mixing maxima from different seasons, a set of GEV parameters is estimated for all maxima from January, another set for all maxima from February and so on. For a given month, GEV parameters are estimated for various durations, e.g. $d \in \{1\,\text{hour}, 6\,\text{hours}, 12\,\text{hours}, 24\,\text{hours}, 48\,\text{hours}, \dots\}$. For a specific return period $T = 1/(1-p)$, with $p$ denoting the non-exceedance probability, a parametric model can be fitted to the corresponding $p$-quantiles $Q_{p,d}$ from GEV distributions for different durations $d$ (e.g., Koutsoyiannis et al., 1998). This model we call IDF curve $\text{IDF}_T(d)$. The estimated IDF-curve $\text{IDF}_{T_1}(d)$ for return period $T_1$ is independent of the estimate of another curve $\text{IDF}_{T_2}(d)$ with return period $T_2 > T_1$. There is no constraint ensuring $\text{IDF}_{T_2}(d) > \text{IDF}_{T_1}(d)$ for arbitrary durations $d$. For example, for a given duration $d$, the 50-year return level can exceed the 100-year return level. Consequently, this approach easily leads to inconsistent (i.e. crossing) IDF-curves. To overcome these problems and increase robustness in constructing IDF curves, Koutsoyiannis et al. (1998) suggested a duration-dependent scale parameter $\sigma_d$

$$\sigma_d = \frac{\sigma}{(d+\theta)^\eta}, \tag{4}$$

with $\theta$, $\eta$ and $\sigma$ being independent of the duration $d$. The parameter $\eta$ quantifies the slope of the IDF curve in the main region and $\theta$ controls the deviation of the power-law behavior for short durations. Furthermore, location is reparametrised by $\tilde{\mu} = \mu/\sigma_d$ which is now independent of the durations $d$ as well as of the shape parameter $\xi$. This leads to the following formulation of a duration-dependent GEV distribution:

$$F(x; \tilde{\mu}, \sigma_d, \xi) = \exp\left\{ -\left[1 + \xi\left(\frac{x}{\sigma_d} - \tilde{\mu}\right)\right]^{\frac{-1}{\xi}} \right\}, \tag{5}$$

which allows consistent modelling of rainfall maxima across different durations $d$ using a single distribution at the cost of only two additional parameters. These parameters can be analogously estimated by maximum-likelihood (Soltyk et al., 2014). To avoid local minima when optimising the likelihood, we repeat the optimisation with different sets of initial guesses for the parameters, sampled again according to a Latin-Hypercube scheme. This method of constructing IDF curves is consistent in the sense that curves for different return periods cannot cross. We refer to this approach as the *duration dependent GEV* approach (dd-GEV). During this work the authors developed and published the R-package IDF (Ritschel et al., 2017). The package includes functions for estimating IDF parameters based on the dd-GEV approach given a precipitation time series and plots the resulting IDF curves.

However, the data points for different durations are dependent (as they are derived from the same underlying high-resolution data set by aggregation) and thus the i.i.d. assumptions required for maximum-likelihood estimation is not fulfilled. Consequently, confidence intervals are not readily available from asymptotic theory; they can be estimated by bootstrapping.

## 4 Data

A precipitation time series from the station *Botanical Garden* in Berlin-Dahlem, Berlin, Germany is used as a case study. A tipping-bucket records precipitation amounts at 1-min resolution. For the analysis at hand, a 13 year time series with 1-min resolution from the years 2001-2013 is available. The series is aggregated to durations $d \in \{1\,\mathrm{hour}, 2\,\mathrm{hours}, 3\,\mathrm{hours}, 6\,\mathrm{hours}, 12\,\mathrm{hours}, 24\,\mathrm{hours}, 48\,\mathrm{hours}, 72\,\mathrm{hours}, 96\,\mathrm{hours}\}$ yielding 9 time series with different temporal resolution. IDF parameters are estimated using annual maxima for each month of the year individually using all 9 duration series.

## 5 Results

### 5.1 Estimation of OBL model parameters

Minimising the symmetric objective function (Eq. (2)) yields OBL model parameter estimates individually for every month of the year, shown in Fig. 3 and explicitly given in Tab. 3 in Appendix A. The resulting OBL model parameters are reasonable compared to observed precipitation characteristics: During summer months, we observe very intensive cells ($\hat{\mu}_x$ between 4mm/h and 8mm/h). However, in June and August, storm duration is relatively short ($\hat{\gamma}$ between 0.25/h and 0.35/h) which can be interpreted as short but heavy thunderstorms which are typically observed in this region in summer (Fischer et al., 2017). Vice versa, in winter small intensities and long storm durations correspond to stratiform precipitation patterns, typically dominating the winter precipitation in Germany. The storm generation rate $\lambda$ shows only a minor seasonal variation.

With the OBL model parameter estimates (Tab. 3, App. A) 1000 realisations with the same length as the observations (13 years) are generated. From both, the original precipitation series and the set of simulated time series, we derive a set of statistics for model validation. The first moment $M_1$ – the mean – is very well represented (not shown) as it enters the objective function with weight $w_1 = 100$ compared to weights of 1 for the other statistics. Figure 4 shows the variance for 6-hourly aggregation and the probability of zero rainfall; for all months the 6h-variances of simulated and observed series are in good agreement. This is particularly noteworthy as the 6-hourly aggregation was not used for parameter estimation. Similar to previous studies (e.g., Onof and Wheater, 1994a), the model fails to reproduce the probability of zero rainfall, here for instance shown for the 12-hourly aggregation. The model mainly overestimates it and therefore has shortcomings in the representation of the time distribution of events (Rodriguez-Iturbe et al., 1987; Onof and Wheater, 1994a).

An important aspect for hydrological applications, is the model's ability to reproduce extremes on various temporal scales. This behaviour is investigated in the next section with the construction of IDF curves.

## 5.2 Intensity-Duration-Frequency curves from OBL model simulations

Monthly block-maxima for every month in the year are drawn for various durations (1h, 3h, 6h, 12h, 24h, 48h, 72h, 96h) from the observational time series and 1000 OBL model simulations of the same length. This is the basis for estimating GEV distributions for individual durations, as well as for constructing dd-GEV IDF curves.

IDF curves for Berlin-Dahlem obtained from observation are shown as dotted lines in Fig. 5 for January, April, July and October for the 0.5-quantile (2-year return period, red), 0.9-quantile (10-year return period, green) and the 0.99-quantile (100-year return period, blue).

Analogously, IDF curves are derived from 1000 simulations of the OBL model precipitation series, cf. Sect. 5.1. The coloured shading in Fig. 5 give the range of variability (5% to 95%) for these 1000 curves with the median highlighted as dotted line. Except for January, the curves obtained directly from the observational series can be found within the range of variability of curves derived from the OBL model. The main IDF features from observations are well reproduced by the OBL model: the power-law-like behaviour (straight line in the double-logarithmic representation) in July extending almost across the full range of durations shown, as well as the flattening of the IDF curves for short durations for April and September. The relative differences in IDF curves given in Fig. 11 (Appendix B) suggest a tendency for the OBL model to underestimate extremes, particularly for large return levels and short durations, similar to results found by, e.g. Verhoest et al. (1997) and Cameron et al. (2000).

Figure 6 shows the relative difference

$$\Delta = \frac{\text{dd-GEV}_{\text{OBL}} - \text{dd-GEV}_{\text{obs}}}{\text{dd-GEV}_{\text{obs}}} \cdot 100\% \tag{6}$$

between IDF curves (dd-GEV) derived from the OBL model dd-GEV$_{OBL}$ including the third moment in parameter estimation (red lines) or alternatively using the probability of zero rainfall to calibrate the model (blue lines), and directly from the observational time series dd-GEV$_{obs}$ for July and two

quantiles: a) 0.5 and b) 0.99 . Including the third moment in parameter estimation slightly improves the model extremes for July for all durations and both short and long return periods. Nevertheless, those promising results could not be found for all months (not shown) and thus we cannot conclude that including the third moment in parameter estimation improves extremes in the OBL model in contrast to findings for the Neyman-Scott variant (Cowpertwait, 1998).

We interpret the different behaviour for short durations (flattening vs continuation of the straight line) for summer (July) and the remaining seasons as a result of different mechanisms governing extreme precipitation events: while convective events dominate in summer, frontal and thus more large scale events dominate in the other seasons.

    As an example, we show segments of time series including the maximum observed/simulated

rainfall in July for durations 1h, 6h and 24h as observed (RR$_{obs}$) and simulated (RR$_{OBL}$) in Fig. 7. Parts of the observed and simulated rainfall time series corresponding to the extreme events for the three different durations are shown in the left and right column, respectively. Additionally the middle column shows the simulated storms and cells generating this extreme event in the artificial time series. As an example, we only show one single model simulation. Visual inspection of several other

simulated series support the main features. For all durations, the extremes are a result of a single long-lasting cell with high intensity. In contrast to an analysis based on the random parameter BL model (Verhoest et al., 2010), these cells are neither unrealistically long nor have an unrealistically high intensity.

    For January, IDF curves from observations and OBL model simulations exhibit large discrepan-

cies: for all durations, the 0.99-quantile (100-yr return level) is above the range of variability from the OBL model and the 0.5-quantile (2-yr return level) is below for small durations. This implies, that the shape of the extreme value distribution characterised by the scale $\sigma$ and shape parameter $\xi$ differs between the two cases. This is likely due to the winter-storm Kyrill hitting Germany and Berlin on January 18$^{th}$ and 19$^{th}$ in 2007 (Fink et al., 2009). We suppose that this rare event is not

sufficiently influential to impact OBL model parameter estimation but does affect the extreme value analysis. For the latter only the one maximum value per month is considered. In fact, the shape parameter $\xi$ estimated from the observational time series shows a large value compared to the other months; in contrast, this value is estimated to be around zero from OBL model simulations. The following section investigates this hypothesis by excluding the precipitation events due to Kyrill.

We furthermore find that the OBL model is generally able to reproduce the observed seasonality in IDF parameters, see Fig. 8. For all parameters, the direct estimation (blue) is mostly within the range of variability of the OBL model simulations. For $\hat{\sigma}$, $\hat{\theta}$ and $\hat{\eta}$, the direct estimation (blue line) features a similar seasonal pattern as the median of the OBL model (red line), whereas for $\hat{\xi}$, the

direct estimation is a lot more erratic than the median (red). As the GEV shape parameter is typically difficult to estimate (Coles, 2001), this erratic behaviour is not unexpected and 11 out of 12 months stay within the expected inner 90% range of variability.

### 5.3 Investigation of the impact of a rare extreme event

The convective cold front passage of Kyrill accounted for a maximum intensity of 24.8mm rainfall per hour, whereas the next highest value of the remaining Januaries would be 4.9mm rainfall per hour in 2002 and thus being more than 5 times lower than for Kyrill. We construct another data set without the extreme event due to Kyrill, i.e. without the year 2007. The intention of this experiment is *not* to motivate removal of an "unsuitable" value. We rather want to show that the OBL model is in generally able to reproduce extremes; it is, however, not flexible enough to account for a single event with magnitude far larger than the rest of the time series. Based on the model with parameters estimated from observations with and without the year 2007 (observed), we obtain return periods for the event "Kyrill" for different durations and find this event to be very rare, especially on short-time scales (1-3 hours), see Tab. 1.

For this data set, we estimate the OBL model parameters and simulate again 1000 time series with these new parameters. The simulated time series were also reduced in length by one year, containing 12 years of rainfall in total. From those precipitation time series, we constructed the dd-GEV IDF curves, see Fig. 9 (right). Without the extreme events due to Kyrill, the OBL model performs in January as well as in the other month with respect to reproducing the IDF relations. In particular, the spread between the 0.5-quantile (2-yr return level) and the 0.99-quantile (100-yr) return level is reduced and the absolute values of extreme quantiles as well, cf. Fig. 9, left and right panel. Note the different scales for the intensity-axes.

### 5.4 Comparing dd-GEV IDF curves to individual duration GEV

In the frame of a model-world study, long time series simulated with the OBL model can be used to investigate adequacy of the dd-GEV model conditional on the simulated series. To this end, we compare the resulting IDF-curves to a GEV distribution obtained for various individual durations. The basis is a set of 1000-year simulations with the OBL model with parameters optimised for Berlin-Dahlem. For a series of this length, we expect to obtain quite accurate (low variance) results for both, the dd-GEV IDF curve and the GEV distributions for individual durations. However, sampling uncertainty is quantified by repeatedly estimating the desired quantities from 50 repetitions. The resulting dd-GEV IDF curves are compared to the individual duration GEV distribution in Fig. 10 for January (left) and July (right).

For most durations in January and July, the dd-IDF curves are close to the quantiles of the individual duration GEV distributions. Notable differences appear for small durations and large quantiles (return levels for long return periods); particularly in January the dd-GEV IDF model overestimates

the 10-year and 100-year return levels (duration of 1h), in July, this effect seems to be present as well but smaller in size. This is accompanied by a slight underestimation of the dd-GEV IDF for durations of 2h to 6h in July and 3h to 6h in January, most visible for the 0.99-quantile (100-yr return level). Both effects together suggest that the flattening of the dd-GEV IDF for small durations is not sufficiently well represented. This could be due to deficiencies in the model for the duration dependent scale parameter (Eq. (4)) but might also be a consequence of an inadequate sampling of durations ($d \in \{1h, 6h, 12h, 24h, 48h, \ldots\}$) to be used to estimate the dd-GEV IDF parameters. This is a point for further investigation.

## 6 Discussion and conclusions

The original version of the Bartlett-Lewis rectangular pulse (OBL) model has been optimised for the Berlin-Dahlem precipitation time series. Subsequently IDF curves have been obtained directly from the original series and from simulation with the OBL model. Basis for the IDF curves has been a parametric model for the duration-dependence of the GEV scale parameter which allows a consistent estimation of one single duration-dependent GEV using all duration series simultaneously (dd-GEV IDF curve). Model parameters for the OBL model and the IDF curves have been estimated for all months of the year and seasonality in the parameters is visible. Typical small-scale convective events in summer and large-scale stratiform precipitation patterns in winter are associated with changes in model parameters.

We have shown that the OBL model is able to reproduce empirical statistics used for parameter estimation; Mean, variance and autocovariance of simulated time series are in good agreement with observational values, whereas the probability of zero rainfall is more difficult to capture (cf. Rodriguez-Iturbe et al., 1987; Onof and Wheater, 1994a).

With respect to the first research question posed in the introduction, we have investigated to what extent the OBL model is able to reproduce the intensity-duration relationship found in observations. We have shown that they do reproduce the main features of the IDF curves estimated directly from the original time series. However, a tendency to underestimate return levels associated with long return periods has been observed similar to Onof and Wheater (1993). Including the third moment in parameter estimation has not significantly improved the OBL model's representation of extremes in contrast to findings for the Neyman-Scott variant (Cowpertwait, 1998).

Furthermore, IDF curves for January show a strong discrepancy between the OBL model simulations and the original series. We have hypothesised and have investigated that this is due to the Berlin-Dahlem precipitation series containing an extreme rainfall event associated with the winter-storm *Kyrill* passing over Berlin during January 18[th] and 19[th], 2007. This event has been very rare in the sense, that on short time scales (e.g. 1 hour and 3 hours) such an event is probable to occur only once within a period larger than 1000 years on average. This addresses the second research question:

How are IDF curves affected by very rare extreme events which are unlikely to be reproduced with the OBL model for a reasonably long simulation? Having excluded the year 2007 from the analysis, the aforementioned discrepancy in January has disappeared. We conclude that an extreme event which is rare (return period of 23000 yrs) with respect to the time scales of simulation ($1000 \times 13$ yrs) has the potential to influence the dd-GEV IDF curve, as 1 out of 13 values per duration (i.e. one maximum per year out of a 13 years time series) does change the GEV distribution. However, its potential to influence mean and variance statistics used to estimate OBL model parameters is minor.

The third question addresses the validity of the duration dependent parametric model for the GEV scale parameter which allows a consistent estimation of IDF curves. For a set of long simulations (1000 years) with the OBL model, the comparison of IDF curves with the duration-dependent GEV approach with quantiles from a GEV estimated from individual durations suggests a systematic discrepancy associated with the flattening of the IDF curve for short durations. Quantiles from individual durations are smaller for short durations than in the dd-GEV approach IDF curves, which is a challenge for the latter modeling approach. However, instead of altering the duration dependent formulation of the scale parameter $\sigma_d$ (Eq. (4)), a different sampling strategy for durations $d$ used in the estimation of the dd-GEV parameters might alleviate the problem. This is a topic for further investigation.

We have not found the OBL model producing unrealistically high precipitation amounts, as discussed for the random-$\eta$ model (Verhoest et al., 2010). Nevertheless, improvements in reproducing the observed extreme value statistics (especially large return levels) could be made by adding a constraint between intensity and duration parameters in the model equations, as a previous study showed (Kaczmarska, 2011).

In summary, the OBL model is able to reproduce the general behaviour of extremes across multiple time scales (durations) as represented by IDF curves. Very rare extreme events do not have the potential to change the OBL model parameters but they do effect IDF statistics and consequently modify the previous conclusion for these cases. A duration dependent GEV is a promising approach to obtain consistent IDF curves; its behaviour at small durations needs further investigation.

## Appendix A: OBL model parameters

In the estimation of OBL model parameters we limited the parameter space by using boundary constraints. Lower and upper parameter limits have been set in a physically realistic range, see Tab. 1. For those parameter ranges, numerical optimisation mostly converged into a global minimum. No constraints are applied in the model variant with the third moment implemented in the OF.

Using a Latin-Hypercube approach, we generated 100 different sets of initial guesses for the parameters used in the numerical optimisation of the symmetrised objective function, Eq. (2). The estimation of OBL model parameters proved to be robust and the majority of optimisation runs led

to the same minimum of the objective function which is then assumed to be the global minimum.

Parameter estimates are given in Tab. 3.

## Appendix B: Difference in IDF curves

Figure 11 shows the relative difference, see Eg. 6, between IDF curves (dd-GEV) derived from the OBL model dd-GEV$_{OBL}$ and directly from the observational time series dd-GEV$_{obs}$. In the four panels in Fig. 11, the discrepancies of the OBL model can be seen. Apart from the large discrepancies

in January discussed in Sect. 5.3, the range of variability (colored shadows in Fig. 11) includes also the zero difference line. However, the median over the 1000 OBL model simulations shows a general tendency for the OBL model to underestimate extremes for large return periods (0.99-quantile) by 25-50%. The best agreement is achieved for April.

*Acknowledgements.* The project has been funded by Deutsche Forschungsgemeinschaft (DFG) through grant

CRC 1114.

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

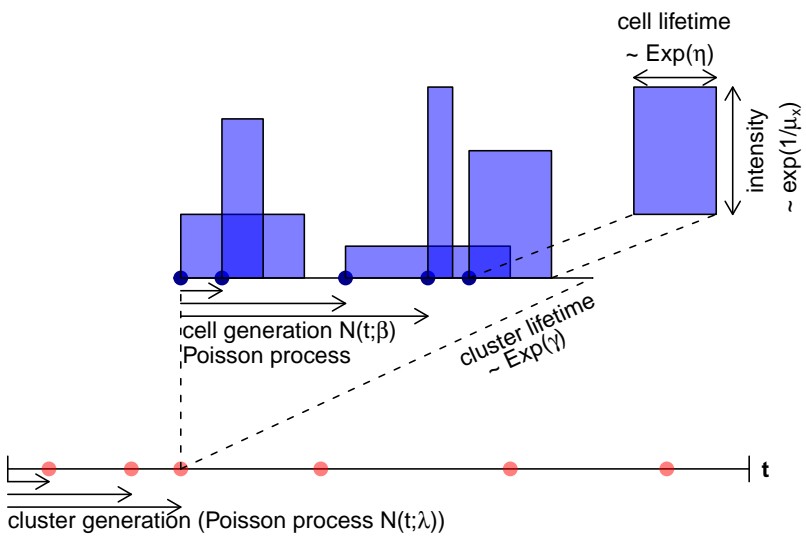

**Figure 1.** Scheme of the OBL model. A similar scheme can be found in Wheater et al. (2005)

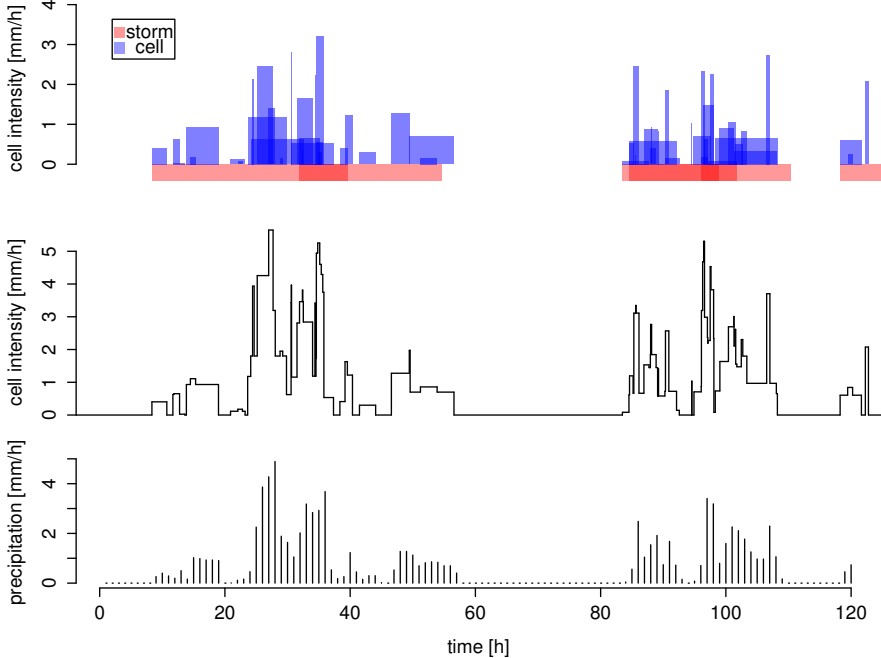

**Figure 2.** Example realisation of the OBL model. The top layer shows the continuously simulated storms and cells by them model. In the middle layer the cell intensities are combined with a step function. The bottom layer shows the aggregated artificial precipitation time series. Used parameters: $\lambda = 4/120\,\mathrm{h}^{-1}$, $\gamma = 1/15\,\mathrm{h}^{-1}$, $\beta = 0.4\,\mathrm{h}^{-1}$, $\eta = 0.5\,\mathrm{h}^{-1}$, $\mu_x = 1\,\mathrm{mm\,h}^{-1}$

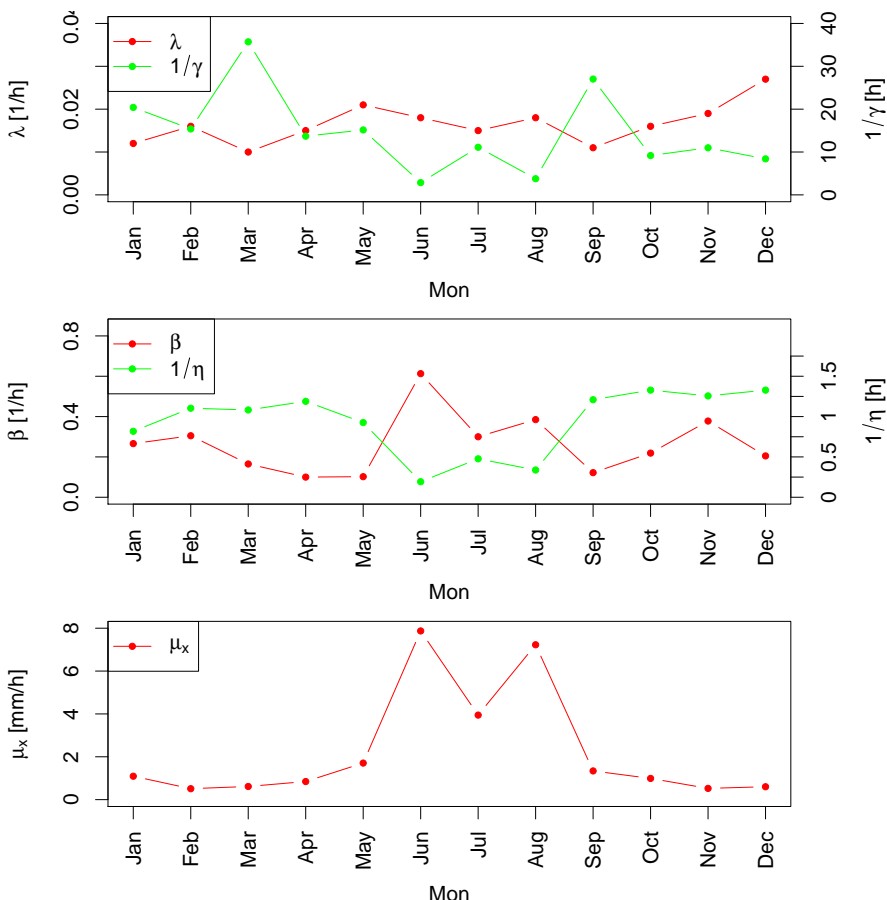

**Figure 3.** OBL model parameter estimates for all months of the year obtained from the Berlin-Dahlem precipitation time series. Top: cell-cluster generation rate $\lambda$ and cluster lifetime $1/\gamma$; middle: cell generation rate $\beta$ and cell lifetime $1/\eta$; bottom: cell mean intensities $\mu_x$.

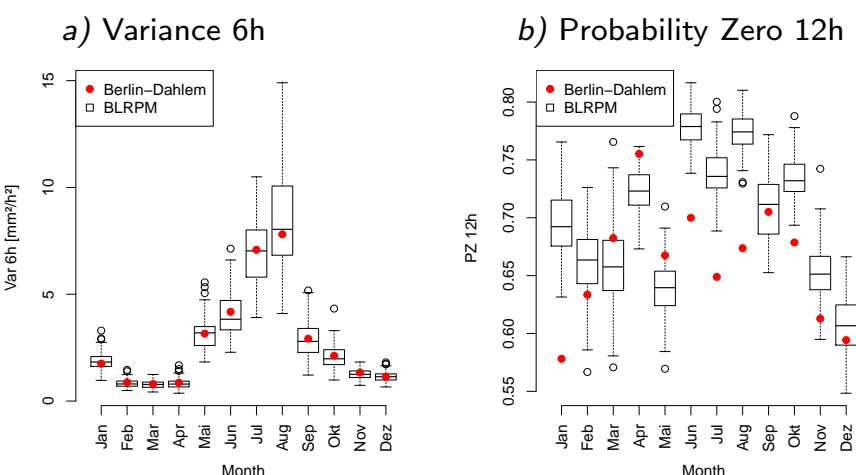

**Figure 4.** Comparison of statistics derived from the observational record (red dots) and 1000 simulated time series (box plots): a) variance at 6-hourly aggregation level and b) probability of zero rainfall at 12-hourly aggregation.

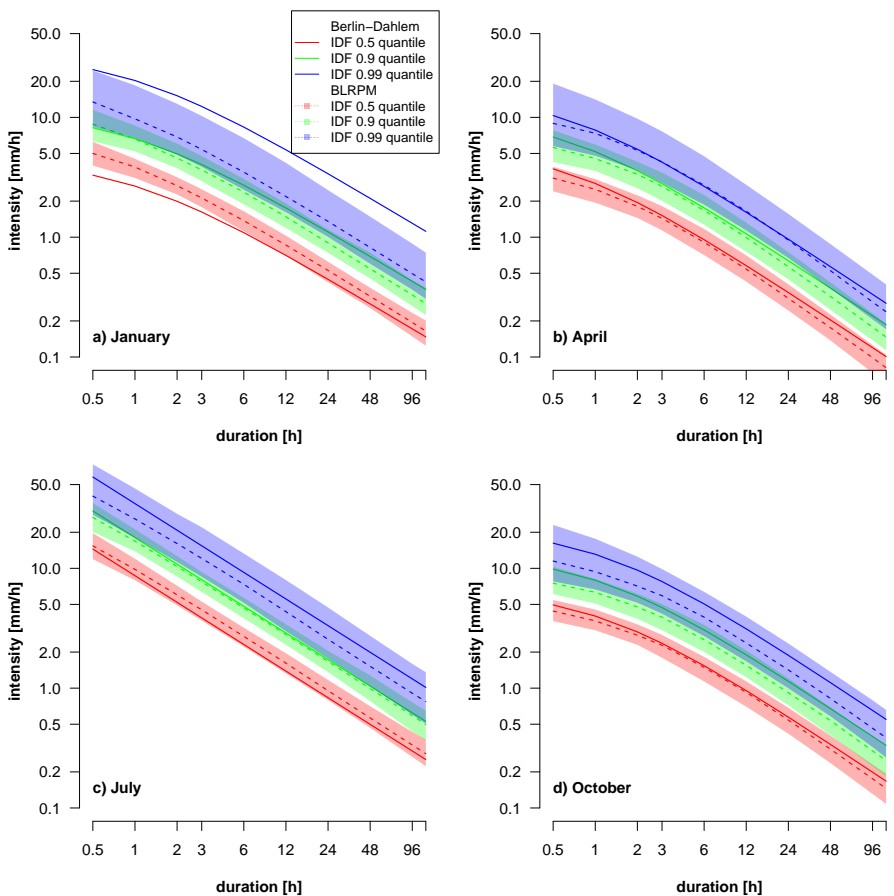

**Figure 5.** IDF curves obtained via dd-GEV for a) January, b) April, c) July and d) October: 0.5 (red), 0.9 (green) and 0.99-quantiles (blue) corresponding to 2-yr, 10-yr, and 100-yr return periods, respectively. Solid lines are derived directly from the Berlin-Dahlem time series. Coloured shadings mark the central 90% range of variability of IDF curves obtained in the same manner with same colour code but from 1000 OBL model simulations (Sect 5.1); the dotted lines mark the median of these curves.

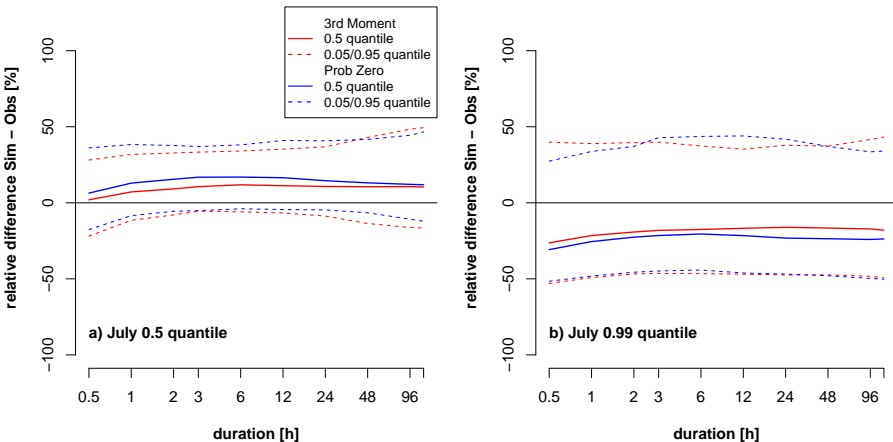

**Figure 6.** Relative differences between observed and simulated return levels obtained with including the third moment (red) and with using the probability of zero rainfall (blue) in parameter estimation for a) July 0.5 quantile and b) July 0.99 quantile. Dotted lines show the 0.05 and 0.95 quantile range of 1000 simulations.

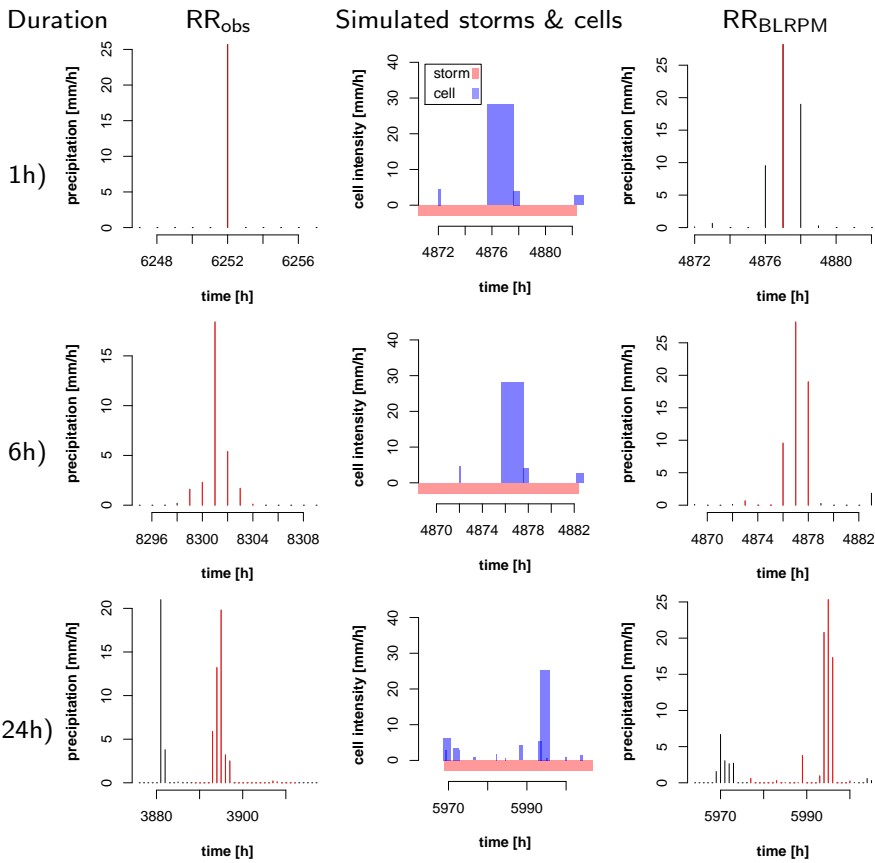

**Figure 7.** Visualisation of July extremes as observed (RR_obs, left column) and simulated by the OBL model (RR_OBL, right column). Shown are short segments including the maximum observed/simulated rainfall (red vertical bars) at durations 1h (top row), 6h (middle row) and 24h (bottom row). Additionally, the middle column shows the simulated storms (red rectangles) and cells (blue rectangles) corresponding to the extreme event of the simulated time series.

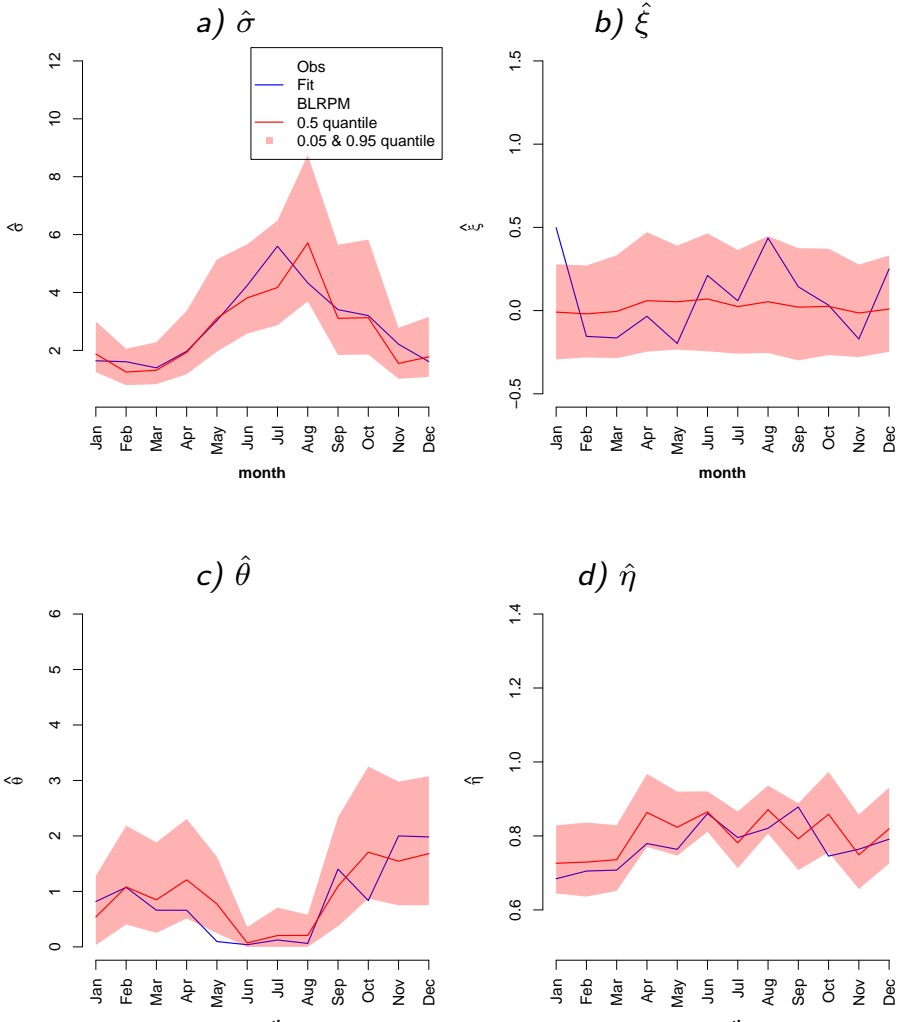

**Figure 8.** Seasonality of IDF model parameters estimated directly from the Berlin-Dahlem series (blue line), and estimated from 1000 OBL model simulations (red). The red shadings give the range of variability (5% to 95%) from the 1000 simulations with the median as solid red line.

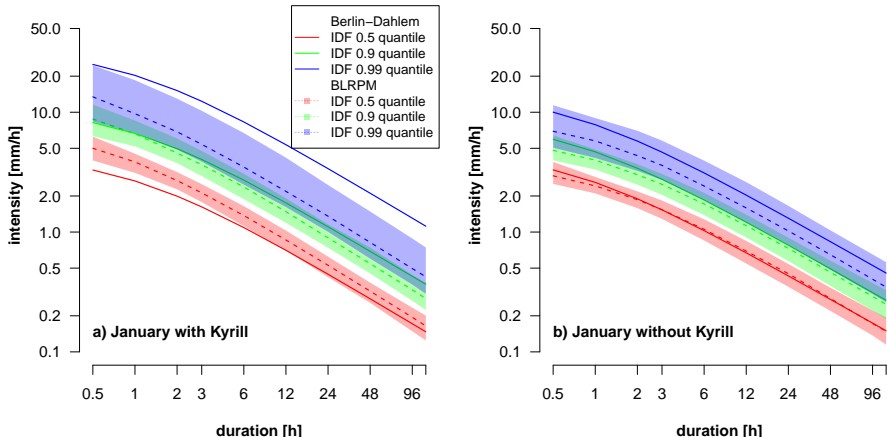

**Figure 9.** dd-GEV IDF curves for a) all Januaries (including 2007), b) Januaries excluding 2007 (different scaling on the intensity axis). Shown are the 0.5 (red), 0.9 (green) and 0.99 (blue) quantile from observations at Berlin-Dahlem (solid lines). The shaded ares are the respective 0.05 and 0.95 quantiles for the associated IDF curves obtained from 1000 OBL model simulations.

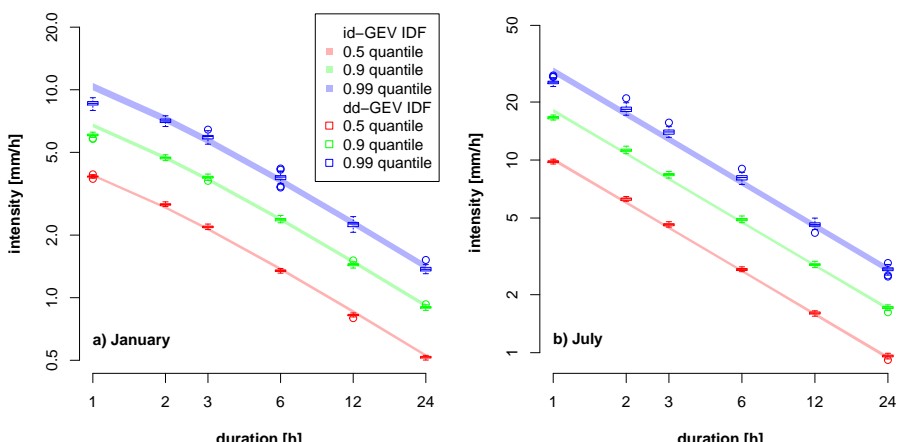

**Figure 10.** dd-GEV IDF curves for a) January and b) July and associated quantiles of a GEV distribution estimated for individual durations. Shown are the 0.5- (red), 0.9- (green) and 0.99-quantile (blue); shaded areas/box plots represent the variability over the 50 repetitions (5% to 95%)

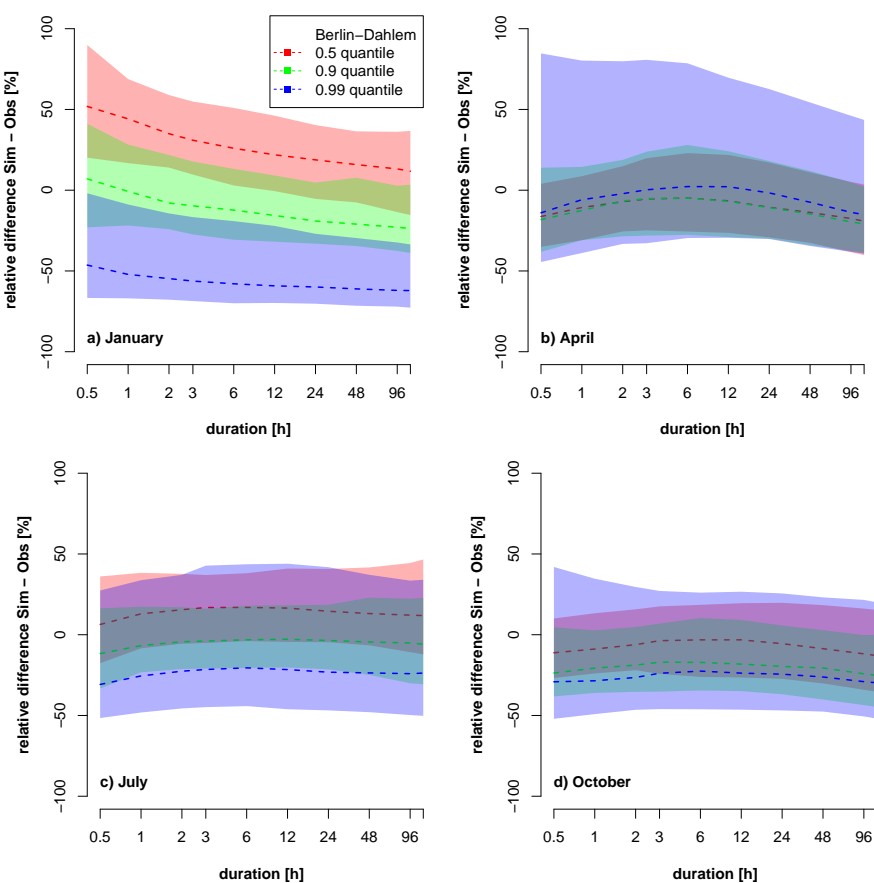

**Figure 11.** Relative differences (Eq. (6)) between simulated and observed IDF curves for a) January, b) April, c) July and d) October in percent relative to the observational values. Shown are the 2-yr (0.5-quantile, red), the 10-yr (0.9-quantile, green) and 100-yr return level (0.99- quantile, blue) differences. The dashed lines denotes the median over all 1000 simulations and the surrounding coloured shading mark the range of variability (5% to 95%). Due to the usage of transparent colours, the three different colours can overlap and mix, grey shadows thus correspond to the overlapping of all three colours.

| Duration [h] | Probability of exceedance without Kyrill [%] | Return period without Kyrill [years] | Probability of exceedance including Kyrill [%] | Return period including Kyrill [years] |
|---|---|---|---|---|
| 1 | $1.8 \times 10^{-6}$ | 560000 | $5.6 \times 10^{-4}$ | 1790 |
| 2 | $4.3 \times 10^{-5}$ | 23000 | $2.4 \times 10^{-3}$ | 420 |
| 3 | $2.2 \times 10^{-4}$ | 4400 | $5.4 \times 10^{-3}$ | 185 |
| 6 | $1.6 \times 10^{-3}$ | 630 | $1.6 \times 10^{-2}$ | 63 |
| 12 | $1.7 \times 10^{-3}$ | 590 | $2.0 \times 10^{-2}$ | 49 |
| 24 | $3.5 \times 10^{-3}$ | 280 | $3.5 \times 10^{-2}$ | 29 |
| 48 | $2.0 \times 10^{-2}$ | 50 | $9.5 \times 10^{-2}$ | 11 |

**Table 1.** Return period for the event *Kyrill* as estimated from the observational time series with this particular event left out and inlcuded for parameter estimation.

| Parameter | Lower boundary | Upper boundary |
|---|---|---|
| $\lambda$ | 0.004 [h$^{-1}$] | 1 [h$^{-1}$] |
| $\gamma$ | 0.01 [h$^{-1}$] | 10 [h$^{-1}$] |
| $\beta$ | 0.01 [h$^{-1}$] | 100 [h$^{-1}$] |
| $\eta$ | 0.01 [h$^{-1}$] | 100 [h$^{-1}$] |
| $\mu_x$ | $1 \times 10^{-9}$ [mm h$^{-1}$] | 100 [mm h$^{-1}$] |

**Table 2.** Boundary constrained used in OBL model parameter estimation.

|  | $\hat{\lambda}\,[\text{h}^{-1}]$ | $\hat{\gamma}\,[\text{h}^{-1}]$ | $\hat{\beta}\,[\text{h}^{-1}]$ | $\hat{\eta}\,[\text{h}^{-1}]$ | $\hat{\mu}_x\,[\text{mm h}^{-1}]$ | $Z_{\text{min}}$ |
|---|---|---|---|---|---|---|
| Jan | 0.012 | 0.049 | 0.266 | 1.223 | 1.093 | 0.389 |
| Feb | 0.016 | 0.065 | 0.305 | 0.906 | 0.511 | 0.036 |
| Mar | 0.010 | 0.028 | 0.165 | 0.924 | 0.614 | 0.077 |
| Apr | 0.015 | 0.073 | 0.100 | 0.841 | 0.845 | 0.125 |
| May | 0.021 | 0.066 | 0.102 | 1.080 | 1.707 | 0.419 |
| Jun | 0.018 | 0.350 | 0.613 | 5.191 | 7.873 | 0.109 |
| Jul | 0.015 | 0.090 | 0.300 | 2.098 | 3.946 | 0.105 |
| Aug | 0.018 | 0.265 | 0.385 | 2.960 | 7.228 | 0.126 |
| Sep | 0.011 | 0.037 | 0.122 | 0.827 | 1.340 | 0.055 |
| Oct | 0.016 | 0.109 | 0.219 | 0.753 | 0.990 | 0.099 |
| Nov | 0.019 | 0.091 | 0.378 | 0.796 | 0.525 | 0.064 |
| Dec | 0.027 | 0.119 | 0.205 | 0.753 | 0.602 | 0.130 |

**Table 3.** Optimum of estimated OBL model parameters for individual months of the year for the Berlin-Dahlem precipitation series and corresponding value of the objective function $Z$