# Peer review of "Precipitation extremes on multiple time scales – Bartlett-Lewis Rectangular Pulse Model and Intensity-Duration-Frequency curves"

_Hydrology and Earth System Sciences, 2017_

## Referee Comment (RC1) · Anonymous Referee #1 · 25 Apr 2017

This paper investigates the ability of the original Bartlett-Lewis model for estimating extreme rainfall at various levels of aggregation. Unfortunately, the paper is not very novel. It is already known for a long period that the Bartlett-Lewis (BL) models have problems in reproducing extremes, especially at shorter aggregation levels. It is not clear why the authors chose for the Original Bartlett-Lewis (OBL) model, while the Modified Bartlett-Lewis (MBL) model or one of the later versions (e.g. Onof and Wheather, 1994) that were further optimized for addressing the problem of the undergeneration of extremes.

An important part of the paper is dealing with the fact that using a short time series for calibration may have an important impact on the statistics described by the observed extremes: the highest extreme may have a much larger return period than the one estimated from the time series. This, of course, is not surprizing, and the shorter the time series used, the higher the potential becomes of facing with extremes that have true return periods much larger than the length of the time series. Yet, this example may be of interest for the scientific community, especially for young researchers starting in the domain of stochastic hydrology. Therefore, I believe this part of the paper may be of interest, though not very novel. Yet, I would like to give some suggestions that may improve this section: (1) using the model with 12 extremes, calculate the return period of the highest extreme that was omitted (i.e. the one in year 2007) to frame how extreme this event in 2007 was? (2) Why not redo the same exercise with the Peak-Over-Threshold method, where the threshold is put quite low to ensure a larger number of extremes? This may reduce the uncertainty on the IDF curves as more data are used to fit the parametric model?

Some minor comments:

Line 1: time series (no dash)

Line 2: data are (the noun "data" is plural)

Line 11-12: here it is not clear what is meant with a singular event. Context is not sufficiently provided.

Lines 33-34: (Koutsoyiannis et al., 1998; Soltyk et al., 2014): braces are put wrongly

Line 73: mention what version of the BL models is used (i.e. the Original BL model)

Line 147: remove the footnote after the equation as it reads as if (1-p) is put to the power "1". The text in the footnote can easily be introduced in the sentence.

Line 181: sentence ends with two dots.

Line 208: put a dot after "length".

Line 227: February is spelled wrongly.

Lines 227-228: please introduce a figure to illustrate this.

Line 232: True, but this is a typical problem occurring for too short time series used for extreme value analysis: fitting a distribution to 13 points is questionable!

Line 243: (Coles, 2001): braces are put wrongly

Line 243: 11 out of 12 months (plural)

Line 289: extent (not extend)

Line 299: "which may not be reproduced by the BLRPM": this may be reproducible! Only, its occurrence may be very low causing that this event was never modelled during the short time series generated! What is the return period of this "singular" event based on the model built from all extremes excluding this event?

Line 330: define "relative difference"

Appendix A: please provide information to the reader of what should be learned from the figures presented in the appendix. Nor the appendix or the text sufficiently elaborates on this.

Figure 5: legend is blurry.

Figure 6: in caption: 1000 simulations: no capital S

Figure 7: text in figure + legend are blurry

Figure 8: legend is blurry

Reference

Onof, C. and Wheater, H. S.: Improvements to the modeling of British rainfall using a modified random parameter Bartlett-Lewis rectangular pulse model, J. Hydrol., 157,

177–195, 1994.

---

## Referee Comment (RC2) · Anonymous Referee #2 · 12 May 2017

Dears

Manuscript title: Precipitation extremes on multiple time scales – Bartlett-Lewis Rectangular Pulse Model and Intensity-Duration-Frequency curves

The comments are included at the bottom of this report.

This paper demonstrates the use of original Bartlett-Lewis models for simulating rainfall series having precipitation extremes on multiple time scales. I believe it is an interesting paper that confirms some of the problems already indicated for the model used. More is needed in terms of discussion and a clearer extreme-value analysis,

possibly involving the examination of other cell intensity distributions and proposed a new version of the model, which they called the Modified Bartlett Lewis (MBL) model.

Please also note the supplement to this comment:
http://www.hydrol-earth-syst-sci-discuss.net/hess-2017-207/hess-2017-207-RC2-supplement.pdf

―――――――――――――――――――

**Supplement:**

Dears

Manuscript title: **Precipitation extremes on multiple time scales –**
**Bartlett-Lewis Rectangular Pulse Model and**
**Intensity-Duration-Frequency curves**

The comments are included at the bottom of this report.

This paper demonstrates the use of original Bartlett-Lewis models for simulating rainfall series having precipitation extremes on multiple time scales. I believe it is an interesting paper that confirms some of the problems already indicated for the model used. More is needed in terms of discussion and a clearer extreme-value analysis, possibly involving the examination of other cell intensity distributions and proposed a new version of the model, which they called the Modified Bartlett Lewis (MBL) model.

The original Bartlett Lewis model is proved efficient to explain the rainfall characteristics at all time intervals considered (1hr to 24hr) as explained by several authors such as Rodriguez-Iturbe et al. (1988) and Onof (1992), a major deficiency is its inability to reproduce the proportion of dry periods correctly. To overcome this problem, Rodriguez-Iturbe et al. (1988) proposed a new version of the model, which they called the Modified Bartlett Lewis (MBL) model. Although several studies have pointed out limitation of the original model and suggested some improvements. Onof and Wheater (1994a), for example, introduced a two- parameter gamma distribution as opposed to the original Bartlett Lewis model which considers a single parameter exponential distribution to describe the depth of a cell in order to better capture extreme events. However, the problem of underestimation of the extreme values still persists, particularly for lower aggregation levels, as described by Verhoest et al.(1997). Vandenberghe et al. (2010) found that the models demonstrated a too severe clustering of rain events.

Comments:

I would recommend the paper to be published after addressing some of the following remarks.

I believe that this work could be improved by better demonstrating the advantages of the original and modified models compared to other rainfall generators (for instance, rectangular pulses models better maintain statistics at different aggregation levels), but also give an overview of drawbacks of the model. For instance, Onof and Wheater (1994) introduced a gamma distribution for the depth of a cell in order to better capture extreme events. Verhoest et al. (2010) discusses that problems still remain as infeasible cells (extremely long) sometimes occur. Vandenberghe et al. (2011) found that the models demonstrated a too severe clustering of rain events. Cameron et al. (2000) and Verhoest et al. (1997) found that these models generally underestimate the extreme values, especially for lower aggregation levels. Onof and Wheater (1993) reported problems for return periods greater than the length of the dataset. According to Cowpertwait (1998) this problem could be overcome if higher order properties would be included in the fitting procedure.

Besides of being in mentioned above, the authors could validate whether the same problems occur for their simulations.

2. Section 2

1. line109: "…the weights, $(w_i; i = 1, 2, ..., k)$ which allow more important weight to be given to fitting some sample moments relative to others. Try to give weights given by $w_i = 1/Var(T_i(y))$ where $Var(T_i(y))$ represents the $i^{th}$ diagonal elements of the covariance matrix of the summary statistics.

2. Give more info on the boundary constraints identified for the parameters of original model that contribute to the stability in the parameter estimates. For the original model, the values of $\lambda$ that are only considered ranges from 0.01 to 0.05.

3. Section 5 Results

1. From results listed in Table 1, it is interesting to observe the higher number of storms with high cell intensity and this is contrary to our prior knowledge about less storm arrivals in dry periods like June. The occurrence of heavy rain in a short duration often induces flush floods in the city area. Form data, it is found the values of cell arrival $\beta$ based on the original model is smaller with high rainfall intensities, particularly for June. This implies that there is a substantial enough cell overlap which could bring extreme rainfall events. Thus, the occurrence of these realistic rainfall cells, whereas, at the hourly time scale, the annual maxima do not generally result from this model.

2. Please check how the extreme events of the original model look like and compare this to the extremes of the historical series. From this you may conclude what is the problem rather than guessing that it has to do with the nature of the rainfall (maybe it is a shortcoming of the model instead! E.g. Verhoest et al. (2010))

---

## Referee Comment (RC3) · Anonymous Referee #3 · 19 May 2017

**Precipitation extremes on multiple time scales – review**

1. The focus on IDF curves as a characteristic of mechanistic models appears to be novel and of wide relevance to hydrological modelling, climate impact assessment and risk estimation. The focus on short duration (5 minute) extremes is also of particular relevance. I therefore think this research is suitable for this publication and would be of general interest to its readership.

2. The paper addresses three research questions which are clearly set out in the introduction. Each question is then addressed in turn in the discussion and conclusions. The questions are as follows:

   I. *"Is the BLRPM able to reproduce the intensity-duration relationship found in observations?"*

   The authors use a depth-dependent GEV distribution (dd-GEV) to estimate extremes across different durations – it is assumed that "across different durations" means "across different temporal scales". Optimisation of the dd-GEV parameters is performed using random sampling from a Latin-Hypercube which appears to be a new method for calibrating these models and is referred to as the depth-dependent GEV approach. This approach is used to construct IDF curves from the observations, and 1000 BLRPM realisations of the same length.

   Typically when we want to estimate extremes from a rainfall model we would sample annual maxima directly from long duration simulations without then using a second extreme value model such as GEV or GP. However, in this case it seems appropriate to apply the dd-GEV for two reasons: 1. to enable direct comparison with the IDF curves from observations, and 2. because the dd-GEV method uses extremes across different scales in fitting. That said, it is not clear from the methodology set out in 5.2 at what scales rainfall has been simulated; is it the same as those used in fitting (i.e., 1, 3, 12, and 24 hrs)? This could be made clearer by the authors.

   The authors note in Section 5.2 (lines 220-2) and in the conclusion (lines 292-3) that the BLRPM tends to under-estimate the extremes. The under-estimation of extremes by mechanistic rainfall models (both Bartlett-Lewis and Neyman-Scott variants), especially at fine temporal scales, is a known issue and the authors' findings are entirely consistent with this. The discussion would be greatly improved by drawing a broader interpretation of the results with comparison with other studies that show under-estimation of extremes by mechanistic models. In particular, is there something to be gained by estimating fine-scale extremes in this way?

   II. *"How are IDF curves affected by a singular extreme event which might not be reproducible with the BLRPM?"*

   BL model parameters are estimated using central moments of the rainfall data therefore it is very likely that this one single extreme will not have as much influence on the estimation of BL model parameters as it does on dd-GEV parameters from observations. And indeed, the authors show that the problem with January disappears when this event is taken out.

   The reader is however left with the impression that the implication is that this event is treated as suspicious information, i.e. that it is fine to take out this largest observation because it is so abnormally larger than any other observed hourly rainfall depth. I don't think that the authors meant this to be the case, but it should be clarified in the text that the section in which this

largest value is taken out does not carry the implication that it is OK to take out the largest value because the event is in some sense 'abnormal'.

This issue brings us to an important problem with the authors' analyses: the data set of 13 years (then reduced to 12 years) is rather short to be doing extreme-value analysis (typically, a peak-over-threshold approach would normally be preferred for such a short dataset. Perhaps the authors' aim is to bring out the greater usefulness of making use of a rainfall model when the data set is not long enough, in which case this should be stated.

III. *"Is the parametric extension of the GEV a valid approach to obtain IDF curves?"*

Here the authors test the validity of the dd-GEV approach to estimating IDF curves by comparing IDF curves obtained from 50 realizations of 1000 years duration from the BL models with GEV estimates from the same simulations. There is an important underlying hypothesis here, namely that the BL model has now been adopted as an accurate representation of the distribution of rainfall (in particular extremes), but we know that this is not true from the problems identified in the analysis of BL's IDF curves. So it is important to qualify the scope of this third research question to make it clear that it is an analysis conditional upon a hypothesis that is only approximately true.

This issue also has a bearing upon the interpretation of the results. For instance, when they identify an under-estimation of 10 and 100 year hourly extremes in January and July, the authors conclude that this is due to poor representation of the dd-GEV IDF curves at these scales which is described as flattening. However, this result is also consistent with the known issue of mechanistic models under-estimating fine-scale (hourly and sub-hourly) extremes yet there is no discussion to this effect.

It is potentially encouraging that the estimation of fine-scale extremes with dd-GEV IDF curves from BL model simulations does not show the underestimation ordinarily obtained from mechanistic models, therefore the authors could explore this in their discussion.

A further issue potentially lies in the estimation of confidence intervals. There may be over-confidence in the extreme value estimates and IDF curves presented in Figure 8. Confidence intervals are estimated from 50 realisations from the BL models. However, GEV extreme value estimates from each realisation would have an associated credible interval which is not shown. It is possible that if this were, then there would be greater overlap in estimation by the two methods and the marginal differences would not be statistically significant.

3. The following are a list of general and specific comments on the analysis presented:
    I. On the whole the English is very good although there are a number of spelling and grammatical errors which should be addressed with a thorough proof-read by the authors prior to final submission. Specific examples can be found on lines 12, 13, 38, and 74 although there may be more. Some errors are flagged in this list.
    II. Line 19: 'affects'
    III. I suggest 'the object of much research' rather than 'the subject of research'
    IV. Line 28: 'in' instead of 'into'
    V. The authors state on lines 44-5 that "Due to the high degree of simplification of the precipitation process, the model is known to have difficulties in the extremes." It is not clear that this is why mechanistic models have a tendency to under-estimate short duration extremes, and many hypotheses have been put forward to address this exact

problem in the literature since their inception in the late 1980s. The authors make a valid point, but it could be enhanced with some references and broader discussion.

VI.  Line 54: add 'of' before 'the BLRPM'

VII.  Line 57: 'estimates'; also, 'ability' rather than 'capability'

VIII.  Line 68: the author's name is 'Le Cam', not 'Cam'. This also needs to be changed in the list of references

IX.  On line 73 the authors highlight that they have chosen to use the original 5 parameter BL model. It would be good to give some justification for using this model variant over the randomised versions of the models, especially given that Kaczmarska, Isham & Onof, (2014) present a new randomised model with enhanced estimation of fine-scale (sub-hourly) extremes.

X.  Add 'a' before 'set'

XI.  On line 87 the authors refer to a "time continuous step function". Should this be "continuous-time"?

XII.  On line 94 the authors comment that the Neyman-Scott model is "…motivated from observations of the distribution of galaxies in space". This sounds fascinating although its relevance to rainfall simulation is perhaps somewhat removed. This statement should be reformulated with an appropriate reference.

XIII.  The sentence on lines 97-9 requires further elaboration.

XIV.  Figure 2:What is the meaning of the red? Is it the duration of the cell generating time (the time during which the storm is active)? And how does it contrast with the blue?

XV.  In Section 2 the authors introduce the BL models and their chosen calibration strategy. On lines 108-10 they highlight their choice of weights with $\omega_i$ = 100 being applied to the first moment $T_i$ (mean). In my experience the mean is usually very well represented by the BL model therefore it is unclear why the authors should want to up-weight this moment so much compared with the others. Given that the authors appear to be using a Generalised Method of Moments, it might be better to weight the summary statistics by the inverse of their observed variance (see http://rsfs.royalsocietypublishing.org/content/1/6/871.figures-only)

XVI.  In lines 123-6 the authors discuss non-identifiability of model parameters although they don't mention if they've checked this for their own calibrations. This could be done by estimating parameter uncertainty or producing profile objective functions on model parameters.

XVII.  Line 151: The equation should read $IDF_{T_2}(d) > IDF_{T_1}(d)$

XVIII.  Line 160: What is meant by 'such a shape parameter $\xi$? Is the claim that $\xi$ is also independent of the scale (duration $d$)? Is that true?

XIX.  It's not clear from the information provided exactly how equation 5 is derived. If this is derived in a previous publication this should be clearly stated and referenced.

XX.  Line 164: It is not clear why there are two extra parameters. It would seem that you are placing several GEV fits (one for each scale) with 3 parameters each, by one fit with 4 parameters (?)

XXI.  In Section 4 it would be useful to identify the gauge resolution. It would also be useful to provide a sentence justifying the choice of gauge location.

XXII.  Line 177: 'records' instead of 'registers'

XXIII.  Line 178: explain why a data set with 13 years only was chosen

XXIV.  Line 200: 'for instance' instead of 'examplary'

XXV.    In Section 5.2, line 210 the authors point the reader to a dotted line in Fig. 5 for IDF curves from observations. In the figure legend, the dotted line is for the IDF curves from BLRPM simulations. This needs to be corrected.

XXVI.   In Section 5.2, line 227 the authors point the reader to February in their discussion of IDF curves in Fig. 5. I think the authors mean January as curves are only presented for January, April, July and October. The authors do the same on line 293 in the conclusions.

XXVII.  Line 233: 'to impact' instead of 'for'

XXVIII. Line 259: add 'to' before 'resulting'

XXIX.   Line 260: 'simulations'

XXX.    Line 282: 'of the year'

XXXI.   Line 289: 'extent'

XXXII.  In the conclusions on lines 314-7 the authors state that they do not find the BLRPM producing unrealistically high precipitation amounts as discussed for the random-η model by Verhoest et al., (2010). The generation of unrealistically high extremes by the modified (random-η) model is specific to that model and is therefore not relevant here as the authors have used the original 5 parameter model.

**References**

Kaczmarska, J., Isham, V. & Onof, C. (2014) Point process models for fine- resolution rainfall. *Hydrological Sciences Journal.* 59 (11), 1972-1991.

Verhoest, N. E., Vandenberghe, S., Cabus, P., Onof, C., Meca-Figueras, T. & Jameleddine, S. (2010) Are stochastic point rainfall models able to preserve extreme flood statistics? *Hydrological Processes.* 24 (23), 3439-3445.

---

## Author Comment (AC1) · 26 Jul 2017

**Precipitation extremes on multiple time scales - Bartlett-Lewis Rectangular Pulse Model and Intensity-Duration-Frequency curves**

—

**Reply to Comments from Reviewer #1**

Christoph Ritschel, Henning Rust, Uwe Ulbrich

July 26, 2017

We are very grateful to three anonymous reviewers for carefully reading and commenting thoroughly on our manuscript. We received highly valuable and constructive comments which very much helped to improve our work and led to new insights. We additionally got plenty ideas for further investigations.

In the following, we go point by point through all the comments and reply to them. Reviewers' comments are all repeated in this document, typeset in black. They are individually addressed, typeset in blue. Changes to the original manuscript as resulting from the reviewers comments are repeated here to ease the comparison with the original version; they are typeset in *blue italic*.

Due to some comments from the reviewers, we decided to exchange the abbreviation *BLRPM* to *OBL model* in order to distinguish the original Bartlett-Lewis model (*OBL*) from a modified version (*MBL*).

**Reviewer #1**

**General Comments:**

This paper investigates the ability of the original Bartlett-Lewis model for estimating extreme rainfall at various levels of aggregation. Unfortunately, the paper is not very novel. It is already known for a long period that the Bartlett-Lewis (BL) models have problems in reproducing extremes, especially at shorter aggregation levels. It is not clear why the authors chose for the Original Bartlett-Lewis (OBL) model, while the Modified Bartlett-Lewis (MBL) model or one of the later versions (e.g. Onof and Wheather, 1994) that were further optimized for addressing the problem of the undergeneration of extremes. An important part of the paper is dealing with the fact that using a short time series for calibration may have an important impact on the statistics described by the observed extremes: the highest extreme may have a much larger return period than the one estimated from the time series. This, of course, is not surprizing, and the shorter the time series used, the higher the potential becomes of facing with extremes that have true return periods much larger than the length of the time series. Yet, this example may be of interest for the scientific community, especially for young researchers starting in the domain of stochastic hydrology. Therefore, I believe this part of the paper may be of interest, though not very novel.

**Remarks:**

Yet, I would like to give some suggestions that may improve this section:

**Mayor (1)**  using the model with 12 extremes, calculate the return period of the highest extreme that was omitted (i.e. the one in year 2007) to frame how extreme this event in 2007 was?

*We added the following sentence and table to Section 5.3: Based on the model with parameters estimated from observations without the year 2007 (observed), we obtain return periods for the event "Kyrill" for different durations and find this event to be very rare, especially on short time scales (1-3 hours), see Tab. 1.*

| Duration [h] | Probability of exceedance without Kyrill [%] | Return period without Kyrill [years] | Probability of exceedance including Kyrill [%] | Return period including Kyrill [years] |
|:---:|:---:|:---:|:---:|:---:|
| 1 | $1.8 \times 10^{-6}$ | 560000 | $5.6 \times 10^{-4}$ | 1790 |
| 2 | $4.3 \times 10^{-5}$ | 23000 | $2.4 \times 10^{-3}$ | 420 |
| 3 | $2.2 \times 10^{-4}$ | 4400 | $5.4 \times 10^{-3}$ | 185 |
| 6 | $1.6 \times 10^{-3}$ | 630 | $1.6 \times 10^{-2}$ | 63 |
| 12 | $1.7 \times 10^{-3}$ | 590 | $2.0 \times 10^{-2}$ | 49 |
| 24 | $3.5 \times 10^{-3}$ | 280 | $3.5 \times 10^{-2}$ | 29 |
| 48 | $2.0 \times 10^{-2}$ | 50 | $9.5 \times 10^{-2}$ | 11 |

Table 1: Return period for the event *Kyrill* as estimated from the observational time series with this particular event left out and included for parameter estimation for different durations.

**Mayor (2)**  Why not redo the same exercise with the Peak-Over-Threshold method, where the threshold is put quite low to ensure a larger number of extremes? This may reduce the uncertainty on the IDF curves as more data are used to fit the parametric model?

*The POT approach might have given us a larger number of extremes. However, we are not sure to what extend the consistent estimation using all durations simultaneously can be performed for the GPD as it can be (and we do it here) for the GEV. Koutsoyiannis et al. [1998] suggested the duration dependent GPD as well as a model for IDF curves but explicitly states that parameter estimation would have to be carried out via annual maxima and the asymptotic equality of GEV and GPD for extremes, rendering the GPD based approach less interesting for us. Our argument here is that uncertainty can be reduced due to borrowing strength from neighbouring durations by using the duration dependent GEV approach.*

**Minor**

- Line 11-12: here it is not clear what is meant with a singular event. Context is not sufficiently provided.

  *Thanks for the hint! Singular is indeed unclear here. We replace occurrences of singular in the text by rare in the sense given in Tab. 1 or . . . rare event (here an event with a return period larger than 1000 years on the hourly time scale) before the table is introduced in section 5.3.*

- **Line 73:** mention what version of the BL models is used (i.e. the Original BL model)

  *We added* original *and* (OBL) *to line* 73 *and change the notation* OBL model *instead of* BLRPM *throughout the text.*

- **Line 147:** remove the footnote after the equation as it reads as if (1-p) is put to the power "1". The text in the footnote can easily be introduced in the sentence.

  *We changed the sentence to* An IDF curve for a given return period $T = 1/(1 - p)$, with $p$ denoting the non-exceedance probability,...

- **Lines 227-228:** please introduce a figure to illustrate this.

  *Thanks for the hint. It should (and does now) read in the text* For January, IDF curves from observations and OBL model simulations ... *and not* February. *The figure for January is provided.*

- **Line 232:** True, but this is a typical problem occurring for too short time series used for extreme value analysis: fitting a distribution to 13 points is questionable!

  *Here we expect that using the simultaneous fit to 9 durations makes this approach more robust. We fit one duration dependent GEV to 117 extreme values (13 years multiplied by 9 durations). They are, however, clearly not independent.*

- **Line 299:** "which may not be reproduced by the BLRPM": this may be reproducible! Only, its occurrence may be very low causing that this event was never modelled during the short time series generated! What is the return period of this "singular" event based on the model built from all extremes excluding this event?

  *From Tab. 1, one can see that the return period for a comparable event (for 2h duration) is several thousand years. However, we do only simulate 1000 years and probabilities of getting such a strong event in this short time period are low. We suggest a better formulation for this sentence in the introduction:* 2) How are IDF curves affected by very rare extreme events which are unlikely to be reproduced with the OBL model for a reasonably long simulation? *and the conclusion* 2) How are IDF curves affected by very rare extreme events which are unlikely to be reproduced with the OBL model for a reasonably long simulation? When the year 2007 is excluded from the analysis, the aforementioned discrepancy in January disappears. We conclude that an extreme event which is rare (return period of 23000 yrs) with respect to the time scales of simulation (1000 × 13 yrs) has the potential to influence the dd-GEV IDF curve as 1 out of 13 values per duration − i.e. one maximum per year out of a 13 years time series − does change the GEV distribution.

- **Line 330:** define "relative difference"

  *To define this term, we changed the beginning of the paragraph to:* Figure 11 shows the relative difference

  $$\Delta = \frac{dd\text{-}GEV_{OBL} - dd\text{-}GEV_{obs}}{dd\text{-}GEV_{obs}} \cdot 100\% \tag{1}$$

  *between IDF curves (dd-GEV) derived from the OBL model dd-GEV$_{OBL}$ and directly from the observational time series dd-GEV$_{obs}$.*

- Appendix A: please provide information to the reader of what should be learned from the figures presented in the appendix. Nor the appendix or the text sufficiently elaborates on this.

Appendix A is referred to twice in the text and gives an overview on estimated OBL model parameters. We consider the information in the table as necessary for reproducible research.

With the Figures in Appendix B, we suggest another way of looking at differences in IDF curves which aims to provide a better understanding of model deficiencies in terms of over- or underestimation of return levels. We changed the sentence referring to Appendix B in Sec. 5.2 to *The relative differences in IDF curves given in Fig. 11 (Appendix B) suggest a tendency for the OBL model to underestimate extremes, particularly for large return levels and short durations, similar to results found by, e.g. Verhoest et al. [1997] and Cameron et al. [2000].*

**References**

D. Cameron, K. Beven, and P. Naden. Flood frequency estimation by continuous simulation under climate change (with uncertainty). *Hydrology and Earth System Sciences Discussions*, 4(3):393–405, 2000.

D. Koutsoyiannis, D. Kozonis, and A. Manetas. A mathematical framework for studying rainfall intensity-duration-frequency relationships. *J. Hydrol.*, 206(1):118–135, 1998.

N. Verhoest, P.A. Troch, and F.P. De Troch. On the applicability of bartlett-lewis rectangular pulse models in the modeling of desing storms at a point. *J. Hydrol.*, 202:108–120, 1997.

---

## Author Comment (AC2) · 26 Jul 2017

**Precipitation extremes on multiple time scales - Bartlett-Lewis Rectangular Pulse Model and Intensity-Duration-Frequency curves**

—

**Reply to Comments from Reviewer #1**

Christoph Ritschel, Henning Rust, Uwe Ulbrich

July 26, 2017

We are very grateful to three anonymous reviewers for carefully reading and commenting thoroughly on our manuscript. We received highly valuable and constructive comments which very much helped to improve our work and led to new insights. We additionally got plenty ideas for further investigations.

In the following, we go point by point through all the comments and reply to them. Reviewers' comments are all repeated in this document, typeset in black. They are individually addressed, typeset in blue. Changes to the original manuscript as resulting from the reviewers comments are repeated here to ease the comparison with the original version; they are typeset in *blue italic*.

Due to some comments from the reviewers, we decided to exchange the abbreviation *BLRPM* to *OBL model* in order to distinguish the original Bartlett-Lewis model (*OBL*) from a modified version (*MBL*).

**Reviewer #2**

**General Comments:**

This paper demonstrates the use of original Bartlett-Lewis models for simulating rainfall series having precipitation extremes on multiple time scales. I believe it is an interesting paper that confirms some of the problems already indicated for the model used. More is needed in terms of discussion and a clearer extreme-value analysis, possibly involving the examination of other cell intensity distributions and proposed a new version of the model, which they called the Modified Bartlett Lewis (MBL) model. The original Bartlett Lewis model is proved efficient to explain the rainfall characteristics at all time intervals considered (1hr to 24hr) as explained by several authors such as Rodriguez-Iturbe et al. (1988) and Onof (1992), a major deficiency is its inability to reproduce the proportion of dry periods correctly. To overcome this problem, Rodriguez-Iturbe et al. (1988) proposed a new version of the model, which they called the Modified Bartlett Lewis (MBL) model. Although several studies have pointed out limitation of the original model and suggested some improvements. Onof and Wheater (1994a), for example, introduced a two-parameter gamma distribution as opposed to the original Bartlett Lewis model which considers a single parameter exponential distribution to describe the depth of a cell in order to better capture extreme events. However, the problem of underestimation of the extreme

values still persists, particularly for lower aggregation levels, as described by Verhoest et al. (1997). Vandenberghe et al. (2010) found that the models demonstrated a too severe clustering of rain events.

**Comments:**

I would recommend the paper to be published after addressing some of the following remarks. I believe that this work could be improved by better demonstrating the advantages of the original and modified models compared to other rainfall generators (for instance, rectangular pulses models better maintain statistics at different aggregation levels), but also give an overview of drawbacks of the model. For instance, Onof and Wheater (1994) introduced a gamma distribution for the depth of a cell in order to better capture extreme events. Verhoest et al. (2010) discusses that problems still remain as infeasible cells (extremely long) sometimes occur. Vandenberghe et al. (2011) found that the models demonstrated a too severe clustering of rain events. Cameron et al. (2000) and Verhoest et al. (1997) found that these models generally underestimate the extreme values, especially for lower aggregation levels. Onof and Wheater (1993) reported problems for return periods greater than the length of the dataset. According to Cowpertwait (1998) this problem could be overcome if higher order properties would be included in the fitting procedure. Besides of being in mentioned above, the authors could validate whether the same problems occur for their simulations.

We are grateful for this comprehensive overview on the deficits associated with the original Bartlett-Lewis model (OBL) and modified versions. We used the OBL to gain an understanding of this type of stochastic precipitation models with the aim to use it in a non-stationary context in future research. Drawbacks of the OBL and also of modified versions are discussed in the literature, as mentioned by the reviewer. These deficits of the OBL might vanish (at least partially) if used in a non-stationary context where model complexity is increased as parameters are linked to large scale flow variables. This is, however, not a point to be discussed here.

Besides gaining experience for our future research plans, the manuscript we presented contributes to a) the analysis of extreme precipitation over a range of time scales in a consistent way using duration-dependent IDF curves (to our knowledge, this has been only briefly touched in Verhoest et al. [1997], and to b) the question whether the duration-dependent GEV is suitable to obtain IDF curves for these kind of models.

In the revised manuscript we introduce a paragraph reporting on the above mentioned issues in section 1:

*Due to the high degree of simplifications of the precipitation process, known drawbacks of the OBL model include the inability to reproduce the proportion dry as reported by Rodriguez-Iturbe et al. [1988] and Onof [1992], and underestimation of extremes as found by, e.g. Verhoest et al. [1997] and Cameron et al. [2000], especially for shorter durations. Furthermore, problems occur for return levels with associated periods longer than the time series used for calibrating the model [Onof and Wheater, 1993]. Several extensions and improvements to the model have been made. Rodriguez-Iturbe et al. [1988] introduced the randomised parameter Bartlett-Lewis model, allowing for different types of cells. Improvements in reproducing the probability of zero rainfall and capturing extremes have been shown for this model [Velghe et al., 1994]. A gamma-distributed intensity parameter and a jitter were introduced by Onof and Wheater [1994b] for more realistic irregular cell intensities. Nevertheless, problems still remain as Verhoest et al. [2010] discussed the occurrence of infeasible (extremely long lasting) cells and a too severe clustering of rain events was found by Vandenberghe et al. [2011]. Including third-order moments in the parameter estimation showed an improvement in the Neyman-Scott models extremes [Cowpertwait, 1998].*

*For the Bartlett-Lewis variant Kaczmarska et al. [2014] found that a randomised parameter model shows no improvement in fit compared to the OBL model for which the skewness was included in the parameter estimation. Furthermore, an inverse dependence between rainfall intensity and cell duration showed improved performance, especially for extremes at short time scales [Kaczmarska et al., 2014]. Here, we focus on the OBL model with and without the third-order moment included. This model is still part of a well-established class of precipitation models and the reduced complexity is appealing as it allows to be used in a non-stationary context [Kaczmarska et al., 2015].*

As mentioned in Section 5.2 of the manuscript, we found the OBL model to underestimate extremes merely for return levels with associated return periods much longer than our observed time series. We report this result now with a reference to Onof and Wheater [1993]. We cannot confirm a significant underestimation associated with short durations as reported by Cameron et al. [2000] and Verhoest et al. [1997], only the tendency is visible in Fig. 11, as is reported in Section 5.2. Small differences are present; we related those, however, to a problem of estimating a consistent IDF for short durations, see Sect. 5.4.

In a 1000 year simulation with the OBL, we could not find any infeasible cells as mentioned by Verhoest et al. [2010]. They discovered the problem for the modified version of the BL model. In our manuscript, we report in the discussion that in our long OBL simulation, this problem does not occur.

Motivated by this reviewer comment, we included the third moment in our objective function, following Cowpertwait [1998] and using the analytical expression derived by Wheater et al. [2006], which – as the reviewer mentions – should overcome some problems of the OBL, see Sect. 2 where we added following paragraph:

*Following studies by Cowpertwait [1998] and Kaczmarska et al. [2014], we include the third moment in the parameter estimation using analytical expressions derived by Wheater et al. [2006], replacing the probability of zero rainfall in the objective function. Thus, still 13 moments are used to calibrate the OBL model. Due to comparability with other studies most of our analyses will not include the third moment though. A comparison between IDF curves of the model calibrated with the third moment and with the probability of zero rainfall will be carried out, to discuss the effect of including the third moment.*

Compared to using the known problematic probability of zero rainfall [Onof and Wheater, 1994a], we could not find a systematic improvement related to extremes. This is discusses in the revised version in section 5.2. as follows:

*Figure 6 shows the relative difference*

$$\Delta = \frac{dd\text{-}GEV_{OBL} - dd\text{-}GEV_{obs}}{dd\text{-}GEV_{obs}} \cdot 100\% \tag{1}$$

*between IDF curves (dd-GEV) derived from the OBL model dd-GEV$_{OBL}$ including the third moment in parameter estimation (red lines) or alternatively using the probability of zero rainfall to calibrate the model (blue lines), and directly from the observational time series dd-GEV$_{obs}$ for July and two quantiles: a) 0.5 and b) 0.99. Including the third moment in parameter estimation slightly improves the model extremes for July for all durations and both short and long return periods. Nevertheless, those promising results could not be found for all months (not shown) and thus we cannot conclude that including the third moment in parameter estimation improves extremes in the OBL model in contrast to findings for the Neyman-Scott variant [Cowpertwait, 1998].*

**Section 2.1)** line 109: ...the weights, $(w_i; i = 1, 2, ..., k)$ which allow more important weight to be given to fitting some sample moments relative to others. Try to give weights given by

[Figure]

Figure 1: Relative differences between observed and simulated return levels obtained with including the third moment (red) and with using the probability of zero rainfall (blue) in parameter estimation for a) July 0.5 quantile and b) July 0.99 quantile. Dotted lines show the 0.05 and 0.95 quantile range of 1000 simulations.

$w_i = 1/Var(T_i(y))$ where $Var(T_i(y))$ represents the $i^{th}$ diagonal elements of the covariance matrix of the summary statistics.

Vanhaute et al. [2012] investigated different objective functions specified in the following (rewritten using a notation consistent with our manuscript):

$$Z(\boldsymbol{\theta}; \mathbf{T}) = \sum_{i=1}^{k} w_i \left[\tau_i(\boldsymbol{\theta}) - T_i\right]^2 \ (OF1)$$

$$Z(\boldsymbol{\theta}; \mathbf{T}) = \sum_{i=1}^{k} \left\{ \left[1 - \frac{\tau_i(\boldsymbol{\theta})}{T_i}\right]^2 + \left[1 - \frac{T_i}{\tau_i(\boldsymbol{\theta})}\right]^2 \right\} \ (OF2) \tag{2}$$

$$Z(\boldsymbol{\theta}; \mathbf{T}) = \sum_{i=1}^{k} 1/\operatorname{Var}[T_i] \left[\tau_i(\boldsymbol{\theta}) - T_i\right]^2 \ (OF3)$$

with the moments $\tau_i(\boldsymbol{\theta})$ derived from model parameters $\boldsymbol{\theta}$ and the empirical moments $T_i$ estimated from the time series.

Here, we use an objective function based on OF2, using a ratio between analytic and empirical moments. In this formulation, first and second order properties are normalised by their characteristic order of magnitudes and are thus comparable. A scaling with variances as suggested by the reviewer is thus not necessary for this particular case. Additionally, we use the weights $w_i$ from OF1 to emphasize the first moment similarly to Cowpertwait et al. [1996], see Sect. 2. We are, however, aware of objective functions like OF1 with weights being the variances of the moments as proposed by the reviewer and also by Kaczmarska et al. [2015] for the non-stationary setting; An approach we plan to pursue in the future.

**Section 2 2)** Give more info on the boundary constraints identified for the parameters of original model that contribute to the stability in the parameter estimates. For the original model,

the values of $\lambda$ that are only considered ranges from 0.01 to 0.05.

Thanks, that is definitely needed for reproducible research. We add the following to Appendix A:
*Estimation of OBL model parameters follow the boundary constraints: For those parameter*

| Parameter | Lower boundary | Upper boundary |
|---|---|---|
| $\lambda$ | $0.004\ [h^{-1}]$ | $1\ [h^{-1}]$ |
| $\gamma$ | $0.01\ [h^{-1}]$ | $10\ [h^{-1}]$ |
| $\beta$ | $0.01\ [h^{-1}]$ | $100\ [h^{-1}]$ |
| $\eta$ | $0.01\ [h^{-1}]$ | $100\ [h^{-1}]$ |
| $\mu_x$ | $1 \times 10^{-9}\ [mm/h]$ | $100\ [mm/h]$ |

Table 1: Boundary constrained used in OBL model parameter estimation.

*ranges, numerical optimisation mostly converged into a global minimum. For the model variant using the third moment in the OF, no constraints are used.*

**Section 5 Results:** **1.** From results listed in Table 1, it is interesting to observe the higher number of storms with high cell intensity and this is contrary to our prior knowledge about less storm arrivals in dry periods like June. The occurrence of heavy rain in a short duration often induces flush floods in the city area. Form data, it is found the values of cell arrival based on the original model is smaller with high rainfall intensities, particularly for June. This implies that there is a substantial enough cell overlap which could bring extreme rainfall events. Thus, the occurrence of these realistic rainfall cells, whereas, at the hourly time scale, the annual maxima do not generally result from this model.

Thank you for pointing us to this interesting observations, we include the following in Section 5.1: *During summer months, we observe very intensive cells ($\hat{\mu}_x$ between 4mm/h and 8mm/h). However, in June and August, storm duration is relatively short ($\hat{\gamma}$ between 0.25/h and 0.35/h) which can be interpreted as short but heavy thunderstorms which are typically observed in this region in summer [Fischer et al., 2017].* This passage replaces following sentences in Section 5.1 in the original manuscript:
Large mean intensities $\hat{\mu}_x$ and short mean cell life-times $1/\hat{\eta}$ in summer correspond to precipitation being dominated by convective events. Similar, the mean cluster life-time $1/\hat{\gamma}$ decreases in summer, whereas the mean cell generation rate $\hat{\beta}$ increases.

**2.** Please check how the extreme events of the original model look like and compare this to the extremes of the historical series. From this you may conclude what is the problem rather than guessing that it has to do with the nature of the rainfall (maybe it is a shortcoming of the model instead! E.g. Verhoest et al. (2010))

We checked extreme events of the OBL model and visually compare them to the extremes of the historical site. We add the following figure and text to the manuscript to section 5.2. *As an example, we show segments of time series including the maximum observed/simulated rainfall in July for durations 1h, 6h and 24h as observed ($RR_{obs}$) and simulated ($RR_{OBL}$) in Fig. 7. Parts of the observed and simulated rainfall time series corresponding to the extreme events for the three different durations are shown in the left and right column, respectively. Additionally the middle column shows the simulated storms and cells generating this extreme event in the*

[Figure]

Figure 2: Visualization of July extremes as observed ($RR_{obs}$, left column) and simulated by the OBL model ($RR_{OBL}$, right column). Shown are short segments including the maximum observed/simulated rainfall (red vertical bars) at durations 1h (top row), 6h (middle row) and 24h (bottom row). Additionally, the middle column shows the simulated storms (red rectangles) and cells (blue rectangles) corresponding to the extreme event of the simulated time series.

*simulated time series. Note, that we show only one simulation as an example; visual inspection*
*of several other simulated series share the main features and are not reproduced here. For all*
*durations, the extremes are a result of a single long-lasting cell with high intensity. In contrast*
*to an analysis based on the random parameter BL model [Verhoest et al., 2010], these cells are*
175 *neither unrealistic long nor have an unrealistic high intensity.*

**References**

D. Cameron, K. Beven, and P. Naden. Flood frequency estimation by continuous simulation under climate change (with uncertainty). *Hydrology and Earth System Sciences Discussions*, 4(3):393–405, 2000.

180 P.S.P. Cowpertwait. A poisson-cluster model of rainfall: high-order moments and extreme values. *Proceedings: Mathematical, Phyisical and Engineering Sciences*, Vol. 454, No. 1971:885–898, 1998.

P.S.P. Cowpertwait, P.E. O'Connell, A.V. Metcalfe, and J.A. Mawdsley. Stochastic point process modelling of rainfall. i. single-site fitting and validation. *J. Hydrol.*, 175:17–46, 1996.

185 M. Fischer, H. W. Rust, and U. Ulbrich. Seasonality in extreme precipitation − using extreme value statistics to describe the annual cycle in german daily precipitation. *Meteorol. Z.*, page accepted, 2017.

J.M. Kaczmarska, V.S. Isham, and C. Onof. Point process models for fine-resolution rainfall. *Hydrological Sciences Journal*, 59(11):1972–1991, 2014.

190 J.M. Kaczmarska, V.S. Isham, and P. Northrop. Local generalised method of moments: an application to point process-based rainfall models. *Environmetrics*, 26:312–325, 2015.

C. Onof. *Stochastic modelling of British rainfall data using Poisson processes*. PhD thesis, University of London, 1992.

C. Onof and H.S. Wheater. Modelling of british rainfall using a random parameter bartlett-lewis 195 rectangular pulse model. *J. Hydrol.*, 149:67–95, 1993.

C. Onof and H.S. Wheater. Improved fitting of the bartlett-lewis rectangular pulse model for hourly rainfall. *Hydrological Sciences*, Vol. 39, No. 6:663–680, 1994a.

C. Onof and H.S. Wheater. Improvements to the modelling of british rainfall using a modified random parameter bartlett-lewis rectangular pulse model. *J. Hydrol.*, 157:177–195, 1994b.

200 I. Rodriguez-Iturbe, D.R. Cox, and V. Isham. A point process model for rainfall: further developments. *Proc. R. Soc. Lond.*, A417:283–298, 1988.

S. Vandenberghe, N. Verhoest, C. Onof, and B. De Baets. A comparative copula-based bivariate frequency analysis of observed and simulated storm events: A case study on bartlett-lewis modeled rainfall. *Water Resources Research*, 47(7), 2011.

205 W. Vanhaute, S. Vandenberghe, K. Scheerlinck, B. De Baets, and N. Verhoest. Calibration of the modified bartlett-lewis model using global optimization techniques and alternative objective functions. *Hydrology and Earth System Sciences*, 16(3):873–891, 2012.

T. Velghe, P.A. Troch, F.P. De Troch, and J. Van de Velde. Evaluation of cluster-based rectangular pulses point process models for rainfall. *Water Resour. Res.*, Vol. 30, No. 10:2847–2857, 1994.

N. Verhoest, P.A. Troch, and F.P. De Troch. On the applicability of bartlett-lewis rectangular pulse models in the modeling of desing storms at a point. *J. Hydrol.*, 202:108–120, 1997.

N. Verhoest, S. Vandenberghe, P. Cabus, C. Onof, T. Meca-Figueras, and S. Jameleddine. Are stochastic point rainfall models able to preserve extreme flood statistics? *Hydrological processes*, 24(23):3439–3445, 2010.

H.S. Wheater, V.S. Isham, R.E. Chandler., C.J. Onof, and E.J. Stewart. Improved methods for national spatial-temporal rainfall and evaporation modelling for bsm. R&D Technical Report F2105/TR, Joint Defra/EA Flood and Coastal Erosion Risk Management R&D Programme, March 2006.

---

## Author Comment (AC3) · 26 Jul 2017

**Precipitation extremes on multiple time scales - Bartlett-Lewis Rectangular Pulse Model and Intensity-Duration-Frequency curves**

—

**Reply to Comments from Reviewer #3**

Christoph Ritschel, Henning Rust, Uwe Ulbrich

July 26, 2017

We are very grateful to three anonymous reviewers for carefully reading and commenting thoroughly on our manuscript. We received highly valuable and constructive comments which very much helped to improve our work and led to new insights. We additionally got plenty ideas for further investigations.

In the following, we go point by point through all the comments and reply to them. Reviewers' comments are all repeated in this document, typeset in black. They are individually addressed, typeset in blue. Changes to the original manuscript as resulting from the reviewers comments are repeated here to ease the comparison with the original version; they are typeset in *blue italic*.

Due to some comments from the reviewers, we decided to exchange the abbreviation *BLRPM* to *OBL model* in order to distinguish the original Bartlett-Lewis model (*OBL*) from a modified version (*MBL*).

**Reviewer #3**

**General Comments**

1. The focus on IDF curves as a characteristic of mechanistic models appears to be novel and of wide relevance to hydrological modelling, climate impact assessment and risk estimation. The focus on short duration (5 minute) extremes is also of particular relevance. I therefore think this research is suitable for this publication and would be of general interest to its readership.
2. The paper addresses three research questions which are clearly set out in the introduction. Each question is then addressed in turn in the discussion and conclusions. The questions are as follows:

**I. "Is the OBL model able to reproduce the intensity-duration relationship found in observations?"** The authors use a depth-dependent GEV distribution (dd-GEV) to estimate extremes across different durations – it is assumed that "across different durations" means "across different temporal scales". Optimisation of the dd-GEV parameters is performed using random sampling from a Latin-Hypercube which appears to be a new method for calibrating these models and is referred to as the depth-dependent GEV approach. This approach is used to construct IDF curves from the observations, and 1000 OBL model realisations of the same length. Typically

when we want to estimate extremes from a rainfall model we would sample annual maxima directly from long duration simulations without then using a second extreme value model such as GEV or GP. However, in this case it seems appropriate to apply the dd-GEV for two reasons: 1. to enable direct comparison with the IDF curves from observations, and 2. because the dd-GEV method uses extremes across different scales in fitting. That said, it is not clear from the methodology set out in 5.2 at what scales rainfall has been simulated; is it the same as those used in fitting (i.e., 1, 3, 12, and 24 hrs)? This could be made clearer by the authors.

As the reviewer wrote, we use a parametric approach to obtain a consistent IDF curve based on a block-maxima approach and a duration-dependent GEV. This idea is based on work by Koutsoyiannis et al. [1998] and later taken up by Soltyk et al. [2014]. The main advantage is to exploit the smoothness in the IDF curve for a more robust estimation. Parameter estimation is carried out by numerically optimising an objective function based on an approximation to the likelihood; the problem of local minima is taken care of by using a latin-hypercube resampling of initial guesses for the parameter optimisation.

From continuous cell simulation, rainfall series have been obtained by aggregating cell rainfall to 1h (minimum duration) and further on to match the duration used for the observed series. We thus include the following sentences at the end of Sect. 2 and augment a sentence at the beginning of Sect. 5.2, respectively

*Simulations with the OBL model are in continuous-time on the level of storms and cells. We aggregate the resulting cell rainfall series to hourly time series.*

*Monthly block-maxima for every month in the year are drawn for various durations (1h, 3h, 6h, 12h, 24h, 48h, 72h, 96h) from the observational time series and 1000 OBL model simulations of same length.*

The authors note in Section 5.2 (lines 220-2) and in the conclusion (lines 292-3) that the OBL model tends to under-estimate the extremes. The under-estimation of extremes by mechanistic rainfall models (both Bartlett-Lewis and Neyman-Scott variants), especially at fine temporal scales, is a known issue and the authors' findings are entirely consistent with this. The discussion would be greatly improved by drawing a broader interpretation of the results with comparison with other studies that show under-estimation of extremes by mechanistic models. In particular, is there something to be gained by estimating fine-scale extremes in this way?

Motivated already by the first comment of Reviewer #2, we related our findings to a broader spectrum of literature, please see our answer in the corresponding document, lines 59-83.

For users, an IDF curve gives a broad and immediate overview about how much (intensity) rain over a period of time (duration) is likely (frequency) to fall. Previous studies mainly focus on Gumbel plots in reference to extreme value analysis. Therefore we believe the presented framework using consistent IDF curves based on a duration-dependent GEV together with stochastic precipitation models can contribute to the community.

**II. "How are IDF curves affected by a singular extreme event which might not be reproducible with the BLRPM?"** BL model parameters are estimated using central moments of the rainfall data therefore it is very likely that this one single extreme will not have as much influence on the estimation of BL model parameters as it does on dd-GEV parameters from observations. And indeed, the authors show that the problem with January disappears when this event is taken out. The reader is however left with the impression that the implication is that this event is treated as suspicious information, i.e. that it is fine to take out this largest

observation because it is so abnormally larger than any other observed hourly rainfall depth. I don't think that the authors meant this to be the case, but it should be clarified in the text that the section in which this largest value is taken out does not carry the implication that it is OK to take out the largest value because the event is in some sense 'abnormal'.

Thanks for pointing this potential problem out! We did not intend to motivate other researchers to take out a "suspicious" date as the winter storm Kyrill in January 2007. Instead we wanted to demonstrate the OBL model's inability to capture characteristics of an event which is much larger in magnitude than the majority of the other events. On the other hand, we showed that the model is generally able to reproduce extreme precipitation events if they are well represented in the underlying data. We augment the first paragraph of Sect. 5.3: *The convective cold front passage of Kyrill accounted for a maximum intensity of 24.8mm rainfall per hour, whereas the next highest value of the remaining Januaries would be 4.9mm rainfall per hour in 2002 and thus being more than 5 times lower than for Kyrill. We construct another data set without the extreme event due to Kyrill, i.e. without the year 2007. The intention of this experiment is not to motivate removal of an "unsuitable" value. We rather want to show that the OBL model is in generally able to reproduce extremes; it is, however, not flexible enough to account for a single event with magnitude far larger than the rest of the time series. . . .*

This issue brings us to an important problem with the authors' analyses: the data set of 13 years (then reduced to 12 years) is rather short to be doing extreme-value analysis (typically, a peak-over-threshold approach would normally be preferred for such a short dataset. Perhaps the authors' aim is to bring out the greater usefulness of making use of a rainfall model when the data set is not long enough, in which case this should be stated.

We admit that an extreme value analysis would benefit from a longer time series, which is unfortunately not available for this case study. With respect to the POT approach, please see our answer to Reviewer #1, Mayor comment 2, lines 40-47 in the corresponding document. It is not our aim to use the OBL model as a relief of the short data series problem. As mentioned in the last comment/answer, we also need a long series to estimate OBL parameters in a way that extremes with a long return period are sufficiently well reproduced.

**III. "Is the parametric extension of the GEV a valid approach to obtain IDF curves?"**
Here the authors test the validity of the dd-GEV approach to estimating IDF curves by comparing IDF curves obtained from 50 realizations of 1000 years duration from the BL models with GEV estimates from the same simulations. There is an important underlying hypothesis here, namely that the BL model has now been adopted as an accurate representation of the distribution of rainfall (in particular extremes), but we know that this is not true from the problems identified in the analysis of BL's IDF curves. So it is important to qualify the scope of this third research question to make it clear that it is an analysis conditional upon a hypothesis that is only approximately true.

Thanks for the hint! Indeed, we do not take the OBL as a representative for the observed rainfall but as a tool to obtain long artificial series to be used in a model-world study. We change the first sentence in Sect. 5.4 to: *In the frame of a model-world study, long time series simulated with the OBL model can be used to investigate adequacy of the dd-GEV model conditional on the simulated series.*

This issue also has a bearing upon the interpretation of the results. For instance, when they identify an under-estimation of 10 and 100 year hourly extremes in January and July, the authors conclude that this is due to poor representation of the dd-GEV IDF curves at these scales which is described as flattening. However, this result is also consistent with the known issue of mechanistic models under-estimating fine-scale (hourly and sub-hourly) extremes yet there is no discussion to this effect. It is potentially encouraging that the estimation of fine-scale extremes with dd-GEV IDF curves from BL model simulations does not show the underestimation ordinarily obtained from mechanistic models, therefore the authors could explore this in their discussion.

Please note, that we now take a single fixed set of parameters to simulate 50 very long (1000yrs) series of rainfall surrogates. Based on these series, we compare two strategies for estimating return levels for different durations: the duration-dependent GEV (dd-GEV) and individual duration GEV approach. Problems of the OBL to represent observed extremes do not play a role here. However, we suggest that the observed effect for short durations indeed needs to be explored in a further analysis.

A further issue potentially lies in the estimation of confidence intervals. There may be over-confidence in the extreme value estimates and IDF curves presented in Figure 8. Confidence intervals are estimated from 50 realisations from the BL models. However, GEV extreme value estimates from each realisation would have an associated credible interval which is not shown. It is possible that if this were, then there would be greater overlap in estimation by the two methods and the marginal differences would not be statistically significant.

Here, we assume that the reviewer uses the term "credible intervals" for the statistical uncertainty intervals, typically associated with any estimator, e.g., here for estimated GEV parameters (or the return levels derived from them). These intervals represent sampling uncertainty, i.e. the uncertainty due to having a particular sample and not the full population available. These estimates can and will vary if another – equally likely but different – sample had been observed. It is exactly this effect which we cover with presenting various samples – i.e. various pseudo-observations – to the GEV estimator. We can do so only in this model-world experiment where we have the model to generate these series. This way of presenting sampling uncertainty is equivalent (at least in interpretation) to the uncertainty intervals based on asymptotic properties of the maximum-likelihood estimator. The latter are typically associated with the GEV or other estimators. However, these asymptotic properties do not hold for the dd-GEV approach and we need a different approach to quantify sampling uncertainty: in this model-world study, we have the possibility to obtain more than one sample and can thus estimate the sampling uncertainty directly from different samples.

**Specific Comments:**

- V. The authors state on lines 44-5 that "Due to the high degree of simplification of the precipitation process, the model is known to have difficulties in the extremes." It is not clear that this is why mechanistic models have a tendency to under-estimate short duration extremes, and many hypotheses have been put forward to address this exact problem in the literature since their inception in the late 1980s. The authors make a valid point, but it could be enhanced with some references and broader discussion.

 References will be included in the revised manuscript as discussed in an answer to Reviewer #2, please see the corresponding document, lines 59-83.

- IX. On line 73 the authors highlight that they have chosen to use the original 5 parameter BL model. It would be good to give some justification for using this model variant over the randomised versions of the models, especially given that Kaczmarska, Isham & Onof, (2014) present a new randomised model with enhanced estimation of fine-scale (sub-hourly) extremes.

    We used the original BL model to gain an understanding of this type of stochastic precipitation models as we plan to use it in a non-stationary setting Kaczmarska et al. [2015], please see also our answer to Reviewer #2, first comment, lines 47-58.

- XI. On line 87 the authors refer to a "time continuous step function". Should this be "continuous-time"?

    Thanks, changed.

- XII. On line 94 the authors comment that the Neyman-Scott model is "...motivated from observations of the distribution of galaxies in space". This sounds fascinating although its relevance to rainfall simulation is perhaps somewhat removed. This statement should be reformulated with an appropriate reference.

    Neyman and Scott developed a model to represent galaxies that tend to cluster. Later the very same model was found to be useful in other contexts, such as rainfall. We decide to leave this original reference in the text as it shows the origins of this model. References to Poisson-cluster models for rainfall are to be found in various places in our manuscript.

- XIII. The sentence on lines 97-9 requires further elaboration.

    We extended this part as follows in the revised manuscript: *Due to known drawbacks of the OBL model several improvements and extensions have been made in the past: Rodriguez-Iturbe et al. [1988] introduced the random parameter model, allowing for different type of cells, and additionally Onof and Wheater [1994] used a jitter and a gamma-distributed intensity parameter to account for a more realistic irregular shape of the cells. Cowpertwait et al. [2007] improved the representation of sub-hourly time scales by adding a third layer, pulses, to the model. Non-stationarity has been addressed by Salim and Pawitan [2003] and Kaczmarska et al. [2015]. Applications of these kind of models include the implementing of copulas to investigate wet and dry extremes [Vandenberghe et al., 2011, Pham et al., 2013], regionalisation [Cowpertwait et al., 1996a,b, Kim et al., 2013] and accounting for interannual variability [Kim et al., 2014].*

- XIV. Figure 2:What is the meaning of the red? Is it the duration of the cell generating time (the time during which the storm is active)? And how does it contrast with the blue?

    The top part of the figure represents a typical OBL simulation of cell clusters, drawn in red, and cells, drawn in blue. The red color corresponds to the life time of the cell cluster or usually referred to as storm. Hereby the vertical extensions of the storm has no physical meaning and only serves for better illustration. During its life time the storm generates rainfall cells (blue). Horizontally illustrated is the cell's life time and during its life time its constant intensity is illustrated by the vertical extension of the cell.

- XV. In Section 2 the authors introduce the BL models and their chosen calibration strategy. On lines 108-10 they highlight their choice of weights with $w_i = 100$ being applied to the first moment $T_i$ (mean). In my experience the mean is usually very well represented by the BL model therefore it is unclear why the authors should want to up-weight this moment so much compared with the others. Given that the authors appear to be using a Generalised Method of Moments, it might be better to weight the summary statistics by the inverse of their observed variance (see )

   As the same point has been risen by Reviewer #2, we refer to our answer in the corresponding document, lines 122-133.

- XVI. In lines 123-6 the authors discuss non-identifiability of model parameters although they don't mention if they've checked this for their own calibrations. This could be done by estimating parameter uncertainty or producing profile objective functions on model parameters.

   We did check the non-identifiability and came to the conclusion the symmetrised objective function is less likely to lead the optimization algorithm into local minima. In five out of six cases the numerical optimization lead to the same (and likely the global) minimum with same parameter values. To our understanding, profile objective functions would inform about *sampling uncertainty* for the given minimum of the OF.

- XVII. Line 151: The notation should read $\mathrm{IDF}_{T_2}(d) > \mathrm{IDF}_{T_2}(d)$.

   Thanks for the hint!

- XVIII. Line 160: What is meant by 'such a shape parameter ? Is the claim that is also independent of the scale (duration )? Is that true?

   After re-parametrising the GEV parameters to $\tilde{\mu} = \mu/\sigma_d$, $\tilde{\mu}$ and the shape parameter $\xi$ are approximately independent of the duration $d$ [Koutsoyiannis et al., 1998]. Please note, that these are only approximations but in the mentioned study it has been shown, that these approximations seem to be well justified.

- XIX. It's not clear from the information provided exactly how equation 5 is derived. If this is derived in a previous publication this should be clearly stated and referenced.

   Given equations (3) and (4), one can introduce the duration dependent scale parameter $\sigma_d$ into equation (3). It results:

$$F(x; \tilde{\mu}, \sigma_d, \xi) = \exp\left\{-\left[1 + \xi\left(\frac{x}{\sigma_d} - \tilde{\mu}\right)\right]^{\frac{-1}{\xi}}\right\}. \tag{1}$$

   Please note, that in the first version of this manuscript the tilde over $\mu$ was missing. This derivation has been made by Koutsoyiannis et al. [1998] and used, e.g. by Soltyk et al. [2014]. This is mentioned in the manuscript.

- XX. Line 164: It is not clear why there are two extra parameters. It would seem that you are placing several GEV fits (one for each scale) with 3 parameters each, by one fit with 4 parameters (?)

   Introducing the duration-dependent scale parameter $\sigma_d$ into the GEV framework leads

to two additional parameters ($\theta$ and $\eta$) and a total of five parameters. These additional parameters describe the dependence of the scale parameter $\sigma_d$ on the duration $d$. As the reviewer mentions, it is indeed possible with this formulation to estimate the IDF relationships over all durations consistently with one single model. Benefit of this approach is a) consistency, in the sense that different quantiles cannot cross along the duration axis, and b) strength in parameter estimation is borrowed from neighbouring durations.

- XXI. In Section 4 it would be useful to identify the gauge resolution. It would also be useful to provide a sentence justifying the choice of gauge location.

  The gauge resolution is one minute, see Sect. 4. The location is chosen due to its vicinity to our institute and interest in local rainfall characteristics, as well as the easy data availability.

- XXIII. Line 178: explain why a data set with 13 years only was chosen

  As mentioned we were interested in local rainfall characteristics in the vicinity of our workplace. Therefore we chose a time series from our weather station in botanical garden Berlin. Also we were interested in the question if a short time series like this can be used for this kind of studies and if it would be sufficiently long enough to gain information about its extreme value distribution. It is known that long rainfall time series with such a high temporal resolution are sparse and many stations do not have long records and thus it is an interesting problem if extreme value distributions can already be obtained from short series. Thus, this study helps in investigating this issue.

- XXV. In Section 5.2, line 210 the authors point the reader to a dotted line in Fig. 5 for IDF curves from observations. In the figure legend, the dotted line is for the IDF curves from BLRPM simulations. This needs to be corrected.

  Thanks for the hint, this mistake was corrected.

- XXVI. In Section 5.2, line 227 the authors point the reader to February in their discussion of IDF curves in Fig. 5. I think the authors mean January as curves are only presented for January, April, July and October. The authors do the same on line 293 in the conclusions.

  Yes, February was put wrongly here and January was meant. This is corrected in the revised manuscript.

- XXXII. In the conclusions on lines 314-7 the authors state that they do not find the BLRPM producing unrealistically high precipitation amounts as discussed for the random-$\eta$ model by Verhoest et al., (2010). The generation of unrealistically high extremes by the modified (random-$\eta$) model is specific to that model and is therefore not relevant here as the authors have used the original 5 parameter model.

  To our knowledge the occurrence of unrealistically high extremes as mentioned by Verhoest et al. (2010) was never investigated for the OBL model and thus we gave it a check. This point was also raised by Reviewer #2, please see our answer in the corresponding document, lines 90-93 and lines 164-175.

**References**

P.S.P. Cowpertwait, P.E. O'Connell, A.V. Metcalfe, and J.A. Mawdsley. Stochastic point process modelling of rainfall. i. single-site fitting and validation. *J. Hydrol.*, 175:17–46, 1996a.

P.S.P. Cowpertwait, P.E. O'Connell, A.V. Metcalfe, and J.A. Mawdsley. Stochastic point process modelling of rainfall. ii. regionalisation and disaggregation. *J. Hydrol.*, 175:47–65, 1996b.

P.S.P. Cowpertwait, V. Isham, and C. Onof. Point process models of rainfall: Developments for fine-scale structure. Research Report 277, Departement of Statistical Science, University College London, June 2007.

J.M. Kaczmarska, V.S. Isham, and P. Northrop. Local generalised method of moments: an application to point process-based rainfall models. *Environmetrics*, 26:312–325, 2015.

D. Kim, F. Olivera, H. Cho, and S.A. Scolofsky. Regionalization of the modified bartlett-lewis rectangular pulse stochastic rainfall model. *Terr. Atmos. Ocean Sci.*, Vol.24, No. 3:421–436, 2013.

D. Kim, J. Kim, and Y.-S. Cho. A poisson cluster stochastic rainfall generator that accounts for the interannual variability of rainfall statistics: validation at various geographic locations across the united states. *Journal of Applied Mathematics*, 2014, 2014.

D. Koutsoyiannis, D. Kozonis, and A. Manetas. A mathematical framework for studying rainfall intensity-duration-frequency relationships. *J. Hydrol.*, 206(1):118–135, 1998.

C. Onof and H.S. Wheater. Improvements to the modelling of british rainfall using a modified random parameter bartlett-lewis rectangular pulse model. *J. Hydrol.*, 157:177–195, 1994.

M. T. Pham, W. Vanhaute, S. Vandenberghe, B. De Baets, and N. Verhoest. An assessment of the ability of bartlett–lewis type of rainfall models to reproduce drought statistics. *Hydrology and Earth System Sciences*, 17(12):5167–5183, 2013.

I. Rodriguez-Iturbe, D.R. Cox, and V. Isham. A point process model for rainfall: further developments. *Proc. R. Soc. Lond.*, A417:283–298, 1988.

A. Salim and Y. Pawitan. Extensions of the bartlett-lewis model for rainfall processes. *Statistical Modelling*, 3(2):79–98, 2003.

S. Soltyk, M. Leonard, A. Phatak, E. Lehmann, et al. Statistical modelling of rainfall intensity-frequency-duration curves using regional frequency analysis and bayesian hierarchical modelling. In *Hydrology and Water Resources Symposium 2014*, page 302. Engineers Australia, 2014.

S. Vandenberghe, N. Verhoest, C. Onof, and B. De Baets. A comparative copula-based bivariate frequency analysis of observed and simulated storm events: A case study on bartlett-lewis modeled rainfall. *Water Resources Research*, 47(7), 2011.

---

## Author Response (AR2)

**Precipitation extremes on multiple time scales - Bartlett-Lewis Rectangular Pulse Model and Intensity-Duration-Frequency curves**

—

**Authors Response to Reviewers**

Christoph Ritschel, Uwe Ulbrich, Peter Névir, Henning W. Rust

October 18, 2017

We are very grateful to one anonymous reviewer and Reik Donner for carefully reading and commenting thoroughly on our manuscript. We received highly valuable and constructive comments which very much helped to improve our work.

In the following, we go point by point through all the comments and reply to them. Reviewers' comments are all repeated in this document, typeset in black. They are individually addressed, typeset in blue. Changes to the original manuscript as resulting from the reviewers comments are repeated here to ease the comparison with the original version; they are typeset in *blue italic*.

**Reviewer 1:**

**General Comments:**

The authors have adequately answered the comments of the reviewers, although I still believe a POT approach would have been better in this case. Some minor errors can still be adjusted.

**Remarks**

- "data" is a plural noun...

  Actually, "data" can be both used as plural and singular. In the first review we got a comment to apply the singular form here.

- Lines 146 to 152 and 164 to 168 contain repetitions, please avoid this and only mention the R-package used once.

  Thanks, changed.

- Lines 186 to 193 should be better explained. In this section, the constraint is wrong: before and after the ">" sign, there is twice the same

  Thanks for the hint. We changed the text slightly and shortened the sentences to provide a better understanding of this section, as follows:

*For a specific return period $T = 1/(1-p)$, with $p$ denoting the non-exceedance probability, a parametric model can be fitted to the corresponding p-quantiles $Q_{p,d}$ from GEV distributions for different durations $d$ [e.g., Koutsoyiannis et al., 1998]. This model we call IDF curve $\mathrm{IDF}_T(d)$. The estimated IDF-curve $\mathrm{IDF}_{T_1}(d)$ for return period $T_1$ is independent of the estimate of another curve $\mathrm{IDF}_{T_2}(d)$ with return period $T_2 > T_1$. There is no constraint ensuring $\mathrm{IDF}_{T_2}(d) > \mathrm{IDF}_{T_1}(d)$ for arbitrary durations $d$. For example, for a given duration $d$, the 50-year return level can exceed the 100-year return level. Consequently, this approach easily leads to inconsistent (i.e. crossing) IDF-curves.*

- Lines 278 to 287 + Figure 7: I don't follow this: How relevant is this: it seems strange to me to make such comparison (unless I didn't understand): how can you compare an observation time series with a modelled one? The modelled time series is not a prediction of the observed one. Probably, the authors wish to demonstrate something which I don't quite follow...

  Yes, it is true, that the modelled time series is no prediction and a comparison here might seem strange. With this, we want to highlight how extreme precipitation amounts are generated in the model and if they are visually comparable to extreme events seen in an observed time series. In Figure 7 we see that extreme events in the model are the result of one long lasting cell with high intensity, which can be interpreted as one frontal convective event in Berlins summer from a meteorological point of view. Also other model simulations showed the same reasons for extreme events, especially on short time scales (1 hour, 6 hours). This will not change the fact, that the model is not able to predict the rainfall, only strengthens the fact, that it is able to capture the nature of Berlin's precipitation patterns and thus a study to compare the IDF curves is legit. The question of how the extreme events are generated in the model came up by 'Reviewer 3' in the first review phase, so we looked into it and added it to the manuscript.

- Line 368: what do you mean with "events short time scales"

  We investigated the extreme event "Kyrill" and its return periods on different time scales. With the "events short time scales" we refer to the smaller aggregation times, e.g. 1 hour and 3 hours.

**Reviewer 4:**

**General Comments:**

- Please comment shortly on the specific selection of the weights for the different moments in Eqs. (1) and (2). Why do you specifically choose these values?

  We use the weights $w_i$ from OF1 to emphasize the first moment similarly to Cowpertwait et al. [1996a], see Sect. 2. The mean precipitation sum is prioritised by a factor of 100 against all the other moments. With that we want to make sure the model is well able to reproduce the mean amount of rainfall, which typically a much needed quantity for hydrological modelling. Of course, other combinations can be discussed and looked at in the future.

- Many of the figures have far too tiny labels and should be replotted so that all the axis labels and legends can be properly read.

Thanks for the hint, all figures except Figure 1 & 2 have been replotted.

- Section 5.3: If I understand correctly, "Kyrill" would represent a "dragon king" rather than a "black swan", i.e., an extreme event that would be unprecedented given the (known) distribution of extremes in the considered time series. Maybe it would be worth briefly referring to these (colloquial) terms (but this is just a suggestion)

  Thanks for mentioning these metaphors and you are probably right about which "Kyrill" is. Nevertheless, we will not refer to them in the final manuscript since they have not yet appeared in studies of the community read by the main author and thus might confuse more than help with the understanding.

**Remarks:**

- P.2, l.46: "proportion dry" sounds very sloppy

  Thanks, changed it to *...proportion of dry periods...*

- P.6, l.131: I recommend specifying here which variable "moments" refers to (i.e., moments of precipitation sums).

  Thanks, we adapted your recommendation.

- P.6, l.136: Please clarify if i=1 applies to the mean at a specific aggregation scale or at all of them.

  Thanks, we clarified: *Here, we use the mean at one hour aggregation time,...*

- P.6, l.141: "with Latin Hypercube" is again too sloppy, please rephrase.

  Thanks, we rephrased: *...using the Latin-Hypercube sampling algorithm...*

- P.6, ll.164-165: This information seems to be partially redundant with ll.149-152, please condense.

  Thanks, we condensed this section.

- P.7, l.187 and below: Please do not use the same symbol "T" for the return period as for the empirical moments $T_i$ in Section 2. This might confuse the reader.

  Thanks, we changed the $T$ for the moments to $M$ throughout the manuscript.

- P.10, l.261: Why do you provide this information in an Appendix instead of just here?

  This is not a completely new information, because it is only the difference of the two graphs which can be already seen in Figure 5 and all the interpretation can be made from this Figure already. Since its not a new nor an important information, but only additional for the interested reader we choose to place it in the appendix.

- P.12, ll.283-284: This sentence reads quite odd and should be rephrased.

  Thanks, we rephrased: *As an example, we only show one single model simulation. Visual inspection of several other simulated series support the main features.*

- Section 6: It is very unusual to write the complete conclusions section in present tense, present perfect would be much more appropriate here.

100     Thanks, we adapted the suggestion.

- P.18, ll.374-375: The end of the sentence "does change the GEV distribution" appears grammatically misplaced.

    Thanks, we used parenthesis instead of hyphen to help with understanding here.

- P.18, ll.381-383: Something seems to be wrong with this sentence, please check and rephrase.

105     Thanks, we changed it to: *Quantiles from individual durations are smaller for short durations than in the dd-GEV approach IDF curves, which is a challenge for the latter modeling approach.*

- P.18, ll.396-398: I hardly understand this short paragraph. Please cross-check and rephrase if necessary.

110     Thanks, we reformulated this section as follows: *In the estimation of OBL model parameters we limited the parameter space by using boundary constraints. Lower and upper parameter limits have been set in a physically realistic range, see Tab. 2. For those parameter ranges, numerical optimisation mostly converged into a global minimum. No constraints are applied in the model variant with the third moment implemented in the OF.*

115 - I did not find Tab. 2 being referenced in the text.

    Thanks, Tab. 2 is now being referenced in the Appendix A.

**Marked-up manuscript version**

[revised manuscript text omitted]

B. Merz, F. Elmer, M. Kunz, B. Mühr, K. Schröter, and S. Uhlemann-Elmer. The extreme flood in june 2013 in germany. *La Houille Blanche*, (1):5–10, 2014.

645   J.A. Nelder and R. Mead. A simplex method for function minimization. *Computer J.*, 7 (4): 308–313, 1965.

J Neyman and EL Scott. A theory of the spatial distribution of galaxies. *Astrophys. J.*, 116:144, 1952.

C. Onof. *Stochastic modelling of British rainfall data using Poisson processes*. PhD thesis,
650   University of London, 1992.

C. Onof and H.S. Wheater. Modelling of british rainfall using a random parameter bartlett-lewis rectangular pulse model. *J. Hydrol.*, 149:67–95, 1993.

C. Onof and H.S. Wheater. Improved fitting of the bartlett-lewis rectangular pulse model for hourly rainfall. *Hydrological Sciences*, Vol. 39, No. 6:663–680, 1994a.

655   C. Onof and H.S. Wheater. Improvements to the modelling of british rainfall using a modified random parameter bartlett-lewis rectangular pulse model. *J. Hydrol.*, 157:177–195, 1994b.

C. Onof, R.E. Chandler, A. Kakou, P. Northrop, H.S. Wheater, and V. Isham. Rainfall modelling using poisson-cluster processes: a review of developments. *Stochastic Environmental Research and Risk Assessment*, 14:184–411, 2000.

A. Pattison. Synthesis of hourly rainfall data. *Water Resour. Res.*, 1 (4):489–498, 1956.

M. T. Pham, W. Vanhaute, S. Vandenberghe, B. De Baets, and N. Verhoest. An assessment of the ability of bartlett–lewis type of rainfall models to reproduce drought statistics. *Hydrology and Earth System Sciences*, 17(12):5167–5183, 2013.

R Core Team. *R: A Language and Environment for Statistical Computing*. R Foundation for Statistical Computing, Vienna, Austria, 2016. URL `https://www.R-project.org/`.

Christoph Ritschel. *BLRPM: Stochastic Rainfall Generator Bartlett-Lewis Rectangular Pulse Model*, 2017. R package version 1.0.

Christoph Ritschel, Carola Detring, and Sarah Joedicke. *IDF: Estimation and Plotting of IDF Curves*, 2017. R package version 1.1.

I. Rodriguez-Iturbe, D.R. Cox, F.R.S., and V. Isham. Some models for rainfall based on stochastic point processes. *Proc. R. Soc. Lond.*, A 410:269–288, 1987.

I. Rodriguez-Iturbe, D.R. Cox, and V. Isham. A point process model for rainfall: further developments. *Proc. R. Soc. Lond.*, A417:283–298, 1988.

A. Salim and Y. Pawitan. Extensions of the bartlett-lewis model for rainfall processes. *Statistical Modelling*, 3(2):79–98, 2003.

D.F. Shanno. Conditioning of quasi-newton methods for function minimization. *Mathematics of Computation*, Vol. 24, No. 111:647–656, 1970.

S.P. Simonovic and A. Peck. *Updated rainfall intensity duration frequency curves for the City of London under the changing climate*. Department of Civil and Environmental Engineering, The University of Western Ontario, 2009.

J.C. Smithers, G.G.S. Pegram, and R.E. Schulze. Design rainfall estimation in south africa using bartlett–lewis rectangular pulse rainfall models. *J. Hydrol.*, 258(1):83–99, 2002.

S. Soltyk, M. Leonard, A. Phatak, E. Lehmann, et al. Statistical modelling of rainfall intensity-frequency-duration curves using regional frequency analysis and bayesian hierarchical modelling. In *Hydrology and Water Resources Symposium 2014*, page 302. Engineers Australia, 2014.

S. Vandenberghe, N. Verhoest, C. Onof, and B. De Baets. A comparative copula-based bivariate frequency analysis of observed and simulated storm events: A case study on bartlett-lewis modeled rainfall. *Water Resources Research*, 47(7), 2011.

W. Vanhaute, S. Vandenberghe, K. Scheerlinck, B. De Baets, and N. Verhoest. Calibration of the modified bartlett-lewis model using global optimization techniques and alternative objective functions. *Hydrology and Earth System Sciences*, 16(3):873–891, 2012.

T. Velghe, P.A. Troch, F.P. De Troch, and J. Van de Velde. Evaluation of cluster-based rectangular pulses point process models for rainfall. *Water Resour. Res.*, Vol. 30, No. 10:2847–2857, 1994.

N. Verhoest, P.A. Troch, and F.P. De Troch. On the applicability of bartlett-lewis rectangular pulse models in the modeling of desing storms at a point. *J. Hydrol.*, 202:108–120, 1997.

N. Verhoest, S. Vandenberghe, P. Cabus, C. Onof, T. Meca-Figueras, and S. Jameleddine. Are stochastic point rainfall models able to preserve extreme flood statistics? *Hydrological processes*, 24(23):3439–3445, 2010.

E.C. Waymire, V.K. Gupta, and I. Rodriguez-Iturbe. A spectral theory of raifnall intensity at the meso-$\beta$ scale. *Water Resour. Res.*, 20(10):1453–1465, 1984.

H.S. Wheater, R.E. Chandler, C.J. Onof, V.S. Isham, E. Bellone, C. Yang, D. Lekkas, G. Lourmas, and M.-L. Segond. Spatial-temporal rainfall modemodel for flood risk estimation. *Stoch. Environ. Res. Risk. Assess*, 19:403–416, 2005.

H.S. Wheater, V.S. Isham, R.E. Chandler., C.J. Onof, and E.J. Stewart. Improved methods for national spatial-temporal rainfall and evaporation modelling for bsm. R&D Technical Report F2105/TR, Joint Defra/EA Flood and Coastal Erosion Risk Management R&D Programme, March 2006.